# Beyond Static Pipelines: Learning Dynamic Workflows for Text-to-SQL

**Yihan Wang** [1]  **Peiyu Liu** [2]  **Runyu Chen** [2]  **Wei Xu** [1]

## Abstract

Text-to-SQL has recently achieved impressive progress, yet remains difficult to apply effectively in real-world scenarios. This gap stems from the reliance on single static workflows, fundamentally limiting scalability to out-of-distribution and long-tail scenarios. Instead of requiring users to select suitable methods through extensive experimentation, we attempt to enable systems to adaptively construct workflows at inference time. Through theoretical and empirical analysis, we demonstrate that optimal dynamic policies consistently outperform the best static workflow, with performance gains fundamentally driven by heterogeneity across candidate workflows. Motivated by this, we propose SquRL, a reinforcement learning framework that enhances LLMs' reasoning capability in adaptive workflow construction. We design a rule-based reward function and introduce two effective training mechanisms: *dynamic actor masking* to encourage broader exploration, and *pseudo rewards* to improve training efficiency. Experiments on widely-used Text-to-SQL benchmarks demonstrate that dynamic workflow construction consistently outperforms the best static workflow methods, with especially pronounced gains on complex and out-of-distribution queries. The codes are available at https://github.com/LLM-Cube/SquRL.

## 1. Introduction

Text-to-SQL aims to translate natural language questions into executable SQL queries, enabling non-expert users to interact with relational databases without requiring specialized knowledge (Shi et al., 2025; Hong et al., 2025). Recent advances in large language models (LLMs), particularly in reasoning and code generation (Jiang et al., 2024), have driven the development of increasingly sophisticated agentic systems, pushing performance toward human-level accuracy on many benchmarks (Li et al., 2023c; Yu et al., 2018).

Despite these advances, applying Text-to-SQL in real-world scenarios remains challenging, as most existing methods assumes that one static fixed workflow can generalize across heterogeneous query distributions, fundamentally constraining scalability to out-of-distribution and long-tail scenarios (Hong et al., 2025). For simple queries, complex pipelines introduce unnecessary latency and computational overhead (Zhu et al., 2025). For difficult queries involving multiple challenges, no single method consistently dominates, requiring the integration of complementary reasoning strategies. Although modular systems like Squrve (Wang et al., 2025b) enable flexible workflow construction, they still rely on human-design rather than selecting the suitable workflow adaptively.

Motivated by this, we propose SquRL, a reinforcement learning (RL) framework that enhances LLM's reasoning capabilities in dynamically workflow construction. Instead of committing to single method performance, the SquRL aims to adaptively select effective workflow based on query and database context, thereby enabling stronger generalization to diverse real-world query scenarios. In the following sections, we focus on addressing two critical questions:

**Q1:** *Does there exist a non-trivial performance gap between the optimal static workflow and the optimal dynamic workflow under realistic query distributions?*

Through theoretical analysis, we demonstrate that the optimal dynamic workflow always performs no worse than the best static workflow, with the performance gap determined by the heterogeneity between candidate workflows. We further provide empirical validation, confirming the potential of dynamic policy in improving both efficiency and accuracy.

**Q2:** *Can we learn a robust policy for dynamically selecting the most appropriate workflow to process Text-to-SQL queries effectively?*

RL-based training relies on high-quality training data, which is scarce for real-world Text-to-SQL scenarios. As the action space grows explosively with the number of components, learning a generalized policy from limited and sparse

[1]Renmin University of China [2]University of International Business and Economics. Correspondence to: Peiyu Liu <liu-peiyustu@163.com>.

*Proceedings of the 43rd International Conference on Machine Learning*, Seoul, South Korea. PMLR 306, 2026. Copyright 2026 by the author(s).

execution feedback becomes challenging. Moreover, the delayed rewards introduced by complete workflow execution make it critical to accelerate training without sacrificing accuracy.

In this study, we first introduce a realization pathway for dynamic workflow construction, which can be implemented from existing static methods. To address sparse execution feedback, we design a rule-based reward function and introduce two effective training mechanisms: dynamic actor masking to encourage broader exploration, and pseudo rewards to improve training efficiency. Extensive experiments on multiple benchmarks demonstrate that SquRL consistently outperforms existing state-of-the-art static baselines, with pronounced gains on complex and out-of-distribution queries. These results demonstrate that dynamic policy learning offers a scalable potential that transcends the inherent limitations of manual workflow design for Text-to-SQL.

## 2. Preliminary

### 2.1. Text-to-SQL

The Text-to-SQL task aims to translate a natural language query into an executable SQL statement. Formally, given a natural language question $q$, a database schema $S$, and optional external knowledge $K$ (Lei et al., 2024), the objective is to generate a SQL query:

$$SQL = F(q, S, K \mid M), \tag{1}$$

where $M$ denotes a large language model and $F$ represents the overall reasoning and generation process. Under this setting, existing methods consistently construct multi-component integrated workflows (e.g., schema linking (Wang et al., 2025c), query decomposition (Pourreza & Rafiei, 2024), and SQL refinement (Li et al., 2025b)) to handle specific real-world challenges. However, these methods often degrade on edge cases even within their intended domains, limiting practical generalization across diverse real-world queries.

In this study, we propose an alternative task-oriented paradigm that aims to dynamically construct the most suitable workflow based on task context, leading to adaptive Text-to-SQL systems.

### 2.2. Actors, Templates, and Workflows

To formalize dynamic workflow construction, we introduce three core abstractions: Actor, Template, and Workflow.

**Actor.** An actor is a modular component that implements a specific function within the Text-to-SQL pipeline, such as schema linking, SQL generation, or execution-guided refinement. Following the unified interface proposed in *Squrve* (Wang et al., 2025b), a recent modular Text-to-SQL framework, heterogeneous components from prior methods are abstracted into a small set of high-level actor categories. [1] For example, `parser` denotes a schema linking component, while `optimizer` denotes a SQL refinement component.

**Template.** A template is an abstract workflow skeleton that specifies a sequence or structure of functional roles (*i.e.,* actor types) without binding them to specific implementations. For example, the DIN-SQL (Pourreza & Rafiei, 2023a) workflow consists of schema linking, subquestion decomposition, SQL generation, and SQL refinement, which can be abstracted as the template: [`parser`, `decomposer`, `generator`, `optimizer`]. In fact, the templates capture the high-level reasoning logic of a workflow and can be instantiated with different actor implementations to realize diverse execution behaviors.

**Workflow.** A workflow is a concrete instantiation of a template by assigning specific Actor implementations to each role. Continuing the above example, the DIN-SQL workflow can be instantiated as: [`dinsql-parser`, `dinsql-decomposer`, `dinsql-generator`, `dinsql-optimizer`], which can be executed by the Squrve framework to produce a SQL query.

### 2.3. Dynamic Workflow Construction

Rather than committing to a single static Text-to-SQL pipeline, we view dynamic Text-to-SQL as a *tool-based reasoning problem*, where solving a query amounts to dynamically constructing a task-specific tool workflow that maximizes expected execution accuracy under the underlying query distribution. Formally, let a query $q$ be drawn from an unknown distribution $D$ over the query space $\mathcal{Q}$. For any executable workflow $W$, we define a Bernoulli random variable:

$$Y_W(q) = \mathbb{I}\{W(q, S, K, M) \text{ executes correctly}\}, \tag{2}$$

which captures whether workflow $W$ successfully solves query $q$. Based on this definition, we consider a combinatorial workflow space:

$$\Omega = \{ f_{\text{match}}(T, A) \mid T \in \mathcal{L}_{\text{template}}, \ A \subseteq \mathcal{L}_{\text{actor}} \}, \tag{3}$$

where each workflow is instantiated by assigning actor implementations $A$ to a template $T$. Accordingly, the expected execution accuracy of a workflow $W$ under $D$ is defined as

$$\mathcal{E}(W) = \mathbb{E}_{q \sim D}[Y_W(q)]. \tag{4}$$

---

[1] Here we interpret actors as *functional tools* in a broad sense: reusable modules that execute specific functions with well-defined input–output interfaces.

With the above formulation, an oracle workflow selector can be defined as

$$W^* = \arg \max_{W \in \Omega} \mathcal{E}(W), \tag{5}$$

which represents the optimal workflow that maximizes expected execution accuracy over the true query distribution. However, directly computing $W^*$ is infeasible in practice. This is because the workflow space $\Omega$ is combinatorial in nature, and oracle supervision over the true query distribution is unavailable. Therefore, instead of explicitly solving for $W^*$, we aim to learn a policy $F_{\text{dynamic}}$ that selects workflows conditioned on $(q, S, K)$ from sparse, execution-based feedback, with the goal of approximating the oracle selector.

$$F_{\text{dynamic}} = f(W^* \mid Q, S, K, M). \tag{6}$$

Next, we first analyze the potential performance gains enabled by dynamically constructing tool workflows, and then present a practical approach for approximating the optimal tool-based reasoning policy.

## 3. Static *vs.* Dynamic: A Theoretical and Empirical Analysis

**Theoretical Analysis.** First, we characterize the fundamental advantage of dynamic workflow construction over any static workflow from a theoretical perspective. Specifically, we establish two key results: (i) the optimal dynamic policy attains execution accuracy no worse than the best static workflow; and (ii) the resulting performance gap is governed by the degree of workflow heterogeneity.

To formalize this comparison, let $\Omega = \{W_1, \dots, W_K\}$ to denote a finite set of workflows and let $q \sim \mathcal{D}$ be a query drawn from the underlying distribution. The expected execution accuracy of the optimal static workflow is:

$$\text{EX}_{\text{static}} = \max_{W \in \Omega} \mathbb{E}_{q \sim \mathcal{D}}[Y_W(q)]. \tag{7}$$

While an oracle dynamic selector that chooses the best workflow for each query achieves

$$\text{EX}_{\text{dynamic}} = \mathbb{E}_{q \sim \mathcal{D}}[\max_{W \in \Omega} Y_W(q)]. \tag{8}$$

We define the performance gap as $\Delta = \text{EX}_{\text{dynamic}} - \text{EX}_{\text{static}}$.

**Theorem 3.1.** *For any finite workflow set $\Omega$ and query distribution $D$, we have $EX_{dynamic} \geq EX_{static}$. Moreover, $\Delta = 0$ if and only if there exists a workflow $W^* \in \Omega$ whose success region covers the union of all workflows' success regions almost surely under $\mathcal{D}$.*

The analysis above reveals a fundamental limitation: when workflows exhibit complementary success regions, no static pipeline can match the expected accuracy achievable by dynamic workflow construction. We next empirically verify that the key theoretical conditions underlying this advantage indeed hold in practice.

**Empirical Validation.** To validate Theorem 3.1 and characterize the upper bound of dynamic workflow selection, we conduct an oracle-style evaluation on the SynSQL dataset (Li et al., 2025a). Specifically, for each query, we exhaustively evaluate all candidate workflows and record the fastest workflow that produces a correct execution. Aggregating these per-query optimal choices yields the oracle accuracy and runtime of dynamic selection, which serves as an upper bound baseline. Detailed experimental settings are provided in Appendix C. Based on this setup, we make the following empirical observations.

> ### Main finding for Q1
>
> *The key bottleneck of prior Text-to-SQL systems is the fixed-workflow assumption rather than model capacity, and reasoning over workflows yields a strictly higher accuracy–efficiency upper bound.*

This conclusion is supported by the following empirical observations: ***Observation One*: Upper bound advantage of dynamic selection.** Figure 1 highlights an oracle upper bound of dynamic workflow selection (red star), obtained by selecting, for each query, the fastest workflow that produces a correct execution. This per-sample optimal selection yields an average execution accuracy of **81.5%** while substantially reducing runtime, outperforming any single static workflow by a large margin. This result demonstrates that the advantage of dynamic selection does not come from a particular workflow, but from the ability to adaptively choose different workflows across queries, thereby achieving a strictly higher accuracy–efficiency upper bound than any fixed strategy. ***Observation Two*: Strong workflow heterogeneity.** We also find substantial pairwise differences among workflows, indicating that they exhibit heterogeneous behaviors across queries. This observation provides empirical support for the heterogeneity assumption underlying Theorem 3.1. Additional analysis details are provided in the Appendix C.2.

## 4. Method

Learning a policy over the combinatorial workflow space is challenging due to the prevalence of invalid compositions and the fact that different queries require different workflows. We therefore decompose policy learning into two stages: *supervised fine-tuning* to constrain the policy to executable workflow structures, and *reinforcement learning* to optimize workflow selection based on execution feedback.

### 4.1. Supervised Fine-Tuning

The goal of supervised fine-tuning is to teach the model to generate valid workflow structures. Instead of optimizing

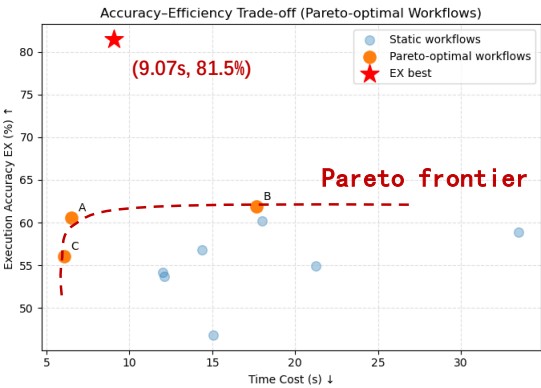

*Figure 1.* Oracle dynamic workflow (red star) *vs.* static workflows.

execution performance, this stage focuses on learning common workflow patterns across queries, covering a range of structural complexity.

#### 4.1.1. TRAINING DATA PREPARATION

We construct supervised training data by transforming standard Text-to-SQL datasets into workflow-level supervision suitable for policy initialization. Specifically, we collect queries from widely used benchmarks, including SynSQL (Li et al., 2025a), Spider (Lei et al., 2024), and BIRD (Li et al., 2023c), which together cover diverse schemas and query complexities. We apply light filtering to remove samples with invalid gold SQL or queries that can be solved by trivial single-step generation, retaining only cases where workflow structure materially affects execution. Since these datasets provide only gold SQL annotations, we derive workflow-level supervision by identifying executable workflows that reproduce the correct execution results. We define a set of workflow templates with increasing structural complexity, ranging from simple single-pass to multi-stage compositions (Appendix C.1). For each query, we progressively explore templates from simpler to more expressive ones and use the first correct workflow as the supervision signal. Queries for which no valid workflow is found are deferred to later optimization. This procedure biases training toward simpler and more efficient workflows whenever sufficient, while still covering queries that require higher-capacity reasoning.

#### 4.1.2. TRAINING OBJECTIVE

Given the constructed workflow-level supervision, we train the model to generate workflows rather than SQL directly. The objective of this stage is to learn a mapping from the query context to a plausible workflow that can produce correct SQL when executed.

Formally, given a query $q$, schema $S$, external knowledge $K$, and the available actor set $\mathcal{A}$, the model generates a

workflow $W$ conditioned on this input using the prompt described in Appendix H:

$$W = \pi_\theta(q, S, K, \mathcal{A}), \tag{9}$$

where $\pi_\theta$ denotes the model parameterized by $\theta$. Unlike prior approaches that directly optimize SQL generation, this objective focuses on learning workflow structures. By doing so, reasoning strategy selection is separated from SQL realization, allowing the model to focus on producing structurally valid and executable workflows at this stage.

### 4.2. Reinforcement Learning

In this section, we further refine workflow selection using *SquRL*, a reinforcement learning framework based on execution feedback. First, we design a rule-based reward that evaluates workflows at multiple points during execution instead of a single binary correctness signal (Section 4.2.1). Even with such a reward, training can still stall when a small number of high-reward workflows dominate the roll-outs. To avoid this, we add randomness during training through dynamic actor masking, which forces the model to explore alternative workflow compositions instead of repeatedly selecting the same pattern (Section 4.2.2). Finally, executing workflows to obtain rewards is costly, especially for complex cases. To keep training efficient, we occasionally replace execution-based rewards with cheaper pseudo rewards, accepting limited noise in exchange for faster feedback (Section 4.2.3).

#### 4.2.1. RULE-BASED REWARD

In practice, reinforcement learning feedback is obtained by executing the SQL produced by each workflow. Instead of relying solely on final execution correctness as in prior work (Yao et al., 2025), we augment this supervision signal by incorporating additional signals from the execution process itself. Specifically, we synthesize the reward from five aspects: *Format Reward*, *Timeout Penalty*, *Execution Reward*, *Result Reward*, and *Time Reward*. These components are applied sequentially in order of increasing execution cost, and evaluation terminates early once a workflow fails at any stage. We next describe each reward component in detail.

**Format Reward.** Before workflow execution, we verify whether the output adheres to the required structural format. Specifically, we require the reasoning trace to be enclosed within `<think>...</think>` tags and the final workflow to be wrapped inside `<answer>...</answer>` using the syntax `list[...]`. Moreover, the generated workflow can only utilize the available actors from the provided actor pool. Based on this, we assign a small reward to

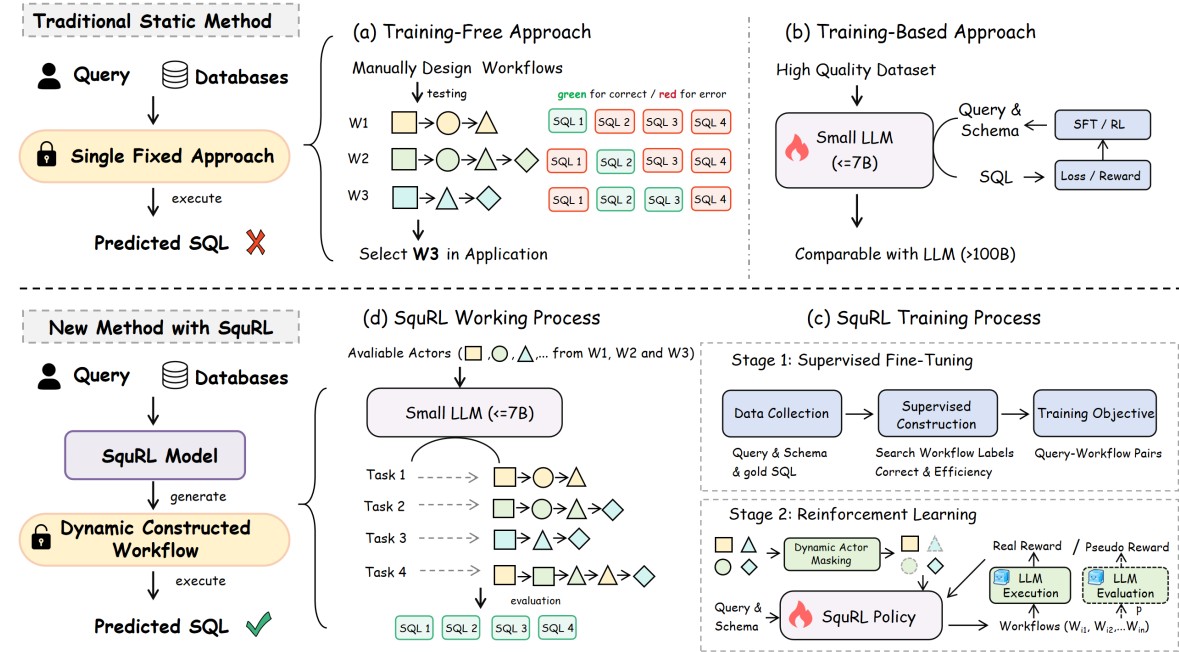

*Figure 2.* Overview of the SquRL framework. Traditional approaches rely on a single fixed workflow to handle diverse query tasks. In contrast, SquRL dynamically constructs workflows tailored to each query, enabling more flexible, accurate, and robust SQL prediction.

encourage the policy to rollout valid workflows.

$$R_f = \begin{cases} +0.5, & \text{if format is correct,} \\ -0.5, & \text{otherwise.} \end{cases} \quad (10)$$

**Timeout penalty.** To avoid training process blocked by erroneous or overly complex workflows, we set a maximum result waiting time for each workflow execution. In practice, we set this timeout parameter to 5 minutes. When execution time exceeds this threshold, we just neglect the final result and assign a negative reward.

$$R_{\text{timeout}} = \begin{cases} -0.5, & \text{if execution timeout,} \\ 0, & \text{otherwise.} \end{cases} \quad (11)$$

**Execution Reward.** Before correctness evaluation, we verify whether workflows generate executable SQL queries. If the workflow generates multiple SQL statements or the generated SQL execution failed due to syntax errors, we assign a negative reward and skip the subsequent process.

$$R_e = \begin{cases} +1, & \text{if generated SQL executable,} \\ -1, & \text{otherwise.} \end{cases} \quad (12)$$

**Result Reward.** Based on comparison with the gold SQL query results, we further evaluate whether the generated SQL returns the required data. We use execution accuracy(EX) metric to measure query correctness. When the execution result is correct, we assign a larger positive reward to the policy.

$$R_r = \begin{cases} +1.5, & \text{if execution result is correct,} \\ -1.5, & \text{otherwise.} \end{cases} \quad (13)$$

**Time Reward.** Conditioned on result correctness, we account for runtime differences among workflows to guide the policy toward more efficient choices.

$$R_t = 0.5 \times \frac{timeout - time}{timeout}, \quad (14)$$

where $time$ denotes the workflow execution time and $timeout$ is as defined previously.

#### 4.2.2. DYNAMIC ACTOR MASKING

To mitigate over-concentration on a small set of high-reward workflows during training, we introduce a dynamic actor masking mechanism. During training, each actor is independently retained with probability $r \in (0,1]$, yielding a reduced actor pool for the current rollout. The model is then restricted to construct workflows using only the retained actors, which effectively perturbs the available action space across rollouts.

Formally, given a masking vector $m$, the workflow space available for a query $q$ is defined as

$$\Omega_q(m) = \{(T, A) \in \Omega \mid A \subseteq \{a_i \mid m_i(q) = 1\}\}, \quad (15)$$

where $m_i(q) = 1$ indicates that actor $a_i$ is available for query $q$.

By randomly perturbing actor availability, this mechanism forces the model to explore alternative workflow compositions instead of repeatedly selecting the same high-reward patterns. When such workflows receive positive feedback, they provide corrective signals that reduce over-reliance on a small set of dominant workflows. In practice, we use a higher retention rate $r$ for more complex queries to increase the likelihood of obtaining informative rewards.

Empirically, we observe that without dynamic masking, training quickly concentrates on a small set of typical workflows, a phenomenon we analyze in detail in Section 5.3.

### 4.2.3. PSEUDO REWARD

To reduce the cost of reward evaluation during reinforcement learning, we introduce a *pseudo reward* mechanism inspired by prior LLM-as-a-judge approaches (Kwon et al., 2023; Gunjal et al., 2025), which replaces a portion of execution-based feedback with LLM-based evaluations. This is particularly important for complex workflows, whose execution often requires substantial time and computational resources.

In practice, we replace full workflow execution with an LLM-based evaluation with a fixed probability $p$. Rather than asking the LLM to judge correctness in isolation, we formulate the evaluation as a pairwise comparison: the LLM is asked to assess whether the current rollout workflow is better than a baseline workflow. This relative comparison provides a more stable and informative signal than absolute judgments. Based on the LLM's output, the pseudo reward is computed as

$$R_{\text{pseudo}} = \begin{cases} 3 + 0.5 \times s, & \text{if the rollout is preferred} \\ -0.5 \times s, & \text{otherwise,} \end{cases} \quad (16)$$

where $s$ denotes the confidence score returned by the LLM. Accordingly, the choice of baseline depends on the availability of ground-truth supervision. If a training sample contains verified ground-truth workflows, we directly use them as baselines. Otherwise, we prompt a stronger LLM to generate a baseline workflow using the same input, which serves as a reference for comparison. In addition, to make the LLM-based evaluation more structured, we summarize a small set of workflow evaluation principles and include them in the judging prompt. These dprinciples are described in Appendix G.

### 4.2.4. TRAINING OBJECTIVE

We adopt Group Relative Policy Optimization (GRPO) (Shao et al., 2024) algorithm for RL-training. Specifically, for each prompt $x$ including the task context $(q, S, K)$ and the available actors $A$, the sampling policy

$\pi_{\theta_{\text{old}}}$ rollouts $G$ candidate responses $\{y_1, y_2, \ldots, y_G\}$, from which workflows $\{W_1, W_2, \ldots, W_G\}$ are parsed. The reward score $R_i$ for each workflow $W_i$ is defined as:

$$R(x, W_i) = \lambda \cdot R_{\text{real}}(x, W_i) + (1-\lambda) \cdot R_{\text{pseudo}}(x, W_i), \quad (17)$$

where $\lambda \in \{0, 1\}$ is a binary indicator with $P(\lambda = 1) = 1 - p$, determining whether the real reward is replaced by a pseudo reward. Then, GRPO computes advantages via group-wise reward normalization:

$$A_i = \frac{R_i - \mu(R_i)}{\sigma(R_i)}. \quad (18)$$

Then, the objective of GRPO is to maximize:

$$\mathcal{L}_{\text{GRPO}}(\theta) = \mathbb{E}_{(x, \{y_i\}_{i=1}^G) \sim \mathcal{D}, \, \pi_{\theta_{\text{old}}}} \left[ \frac{1}{G} \sum_{i=1}^{G} \frac{1}{|y_i|} \sum_{t=1}^{|y_i|} \right.$$
$$\min \left( r_\theta(x, y_i^t) A_i, \ \text{clip}\left(r_\theta(x, y_i^t), 1 - \epsilon, 1 + \epsilon\right) A_i \right)$$
$$\left. - \beta \, \text{KL}(\pi_\theta(\cdot|x) \, \| \, \pi_{\text{ref}}(\cdot|x)) \right], \quad (19)$$

where $r_\theta(x, y_i^t)$ is the importance sampling ratio used to balance the discrepancy between the reasoning engine and training engine under the on-policy setting:

$$r_\theta(x, y_i^{\leq t}) = \frac{\pi_\theta(y_i^t \mid x, y_i^{<t})}{\pi_{\theta_{\text{old}}}(y_i^t \mid x, y_i^{<t})}. \quad (20)$$

## 5. Experiments

### 5.1. Experimental Setup

**Datasets.** We evaluate SquRL on three widely adopted Text-to-SQL benchmarks, Spider (Yu et al., 2018), BIRD (Li et al., 2023c), Spider 2.0 (Lei et al., 2024), and SynSQL (Li et al., 2025a) datasets, which collectively cover a broad range of query difficulty, schema heterogeneity, and domain diversity, enabling comprehensive assessment under real-world settings. Dataset details are provided in Appendix D.

**Baselines.** We compare SquRL with five advanced baselines integrated into Squrve: DIN-SQL (Pourreza & Rafiei, 2023a), CHESS (Talaei et al., 2024), MAC-SQL (Wang et al., 2025a), RSL-SQL (Cao et al., 2024), and LinkAlign (Wang et al., 2025c). These methods employ diverse reasoning strategies, including chain-of-thought (Wei et al., 2022), self-consistency (Wang et al., 2022), and multi-agent collaboration, which exhibit improved performance on complex Text-to-SQL queries. Additionally, we employ actors decoupled from these methods to support dynamic workflow construction.

*Table 1.* Performance comparison across different Text-to-SQL benchmarks. For each query, we generate five workflows and select the final SQL result through majority voting based on the execution results.

| Methods | Spider-Dev | Spider-Test | Bird-Dev | SynSQL | | | | |
|---|---|---|---|---|---|---|---|---|
| | | | | Easy | Moderate | Complex | Highly Complex | All |
| DIN-SQL | 83.14 | 82.07 | 59.62 | 68.11 | 42.19 | 36.21 | 29.57 | 44.17 |
| MAC-SQL | 82.07 | 82.53 | 61.77 | 77.41 | 51.16 | 45.18 | 37.87 | 53.08 |
| LinkAlign | 81.94 | 81.79 | 59.75 | 65.12 | 47.51 | 46.18 | 32.26 | 47.92 |
| RSL-SQL | 84.30 | 83.74 | 65.82 | 78.01 | 61.46 | 50.83 | 38.54 | 57.42 |
| CHESS | 79.20 | 79.56 | 63.10 | 72.51 | 45.08 | 39.35 | 28.90 | 48.03 |
| SquRL-1.5B | 85.10 | 84.83 | 64.47 | 77.08 | 57.81 | 52.16 | 38.87 | 56.67 |
| SquRL-3B | 85.55 | 84.32 | 65.59 | 75.96 | 65.89 | 56.68 | 44.02 | 60.85 |
| SquRL-7B | **86.27** | **86.58** | **67.60** | **82.06** | **68.78** | **59.47** | **46.51** | **64.42** |

**Evaluation Metrics.** Following standard Text-to-SQL practice, we primarily use Execution Accuracy (EX) as our evaluation metric, which assesses SQL correctness by comparing execution results against gold SQL statements. We follow Ma et al. (2025) and intentionally avoid Exact Match metric, since logically equivalent SQL queries may differ syntactically.

**Implementations.** We develop our reinforcement learning framework based on VERL (Sheng et al., 2024). We execute the workflows and evaluate the final SQL results using Squrve as the backend sandbox environment. For policy model, we use Qwen2.5-7B-Instruct [2] as the backbone. For actors that require stronger reasoning, we additionally employ Qwen-Plus via API access.

## 5.2. Main Results

Table 1 summarizes the main experimental results, comparing SquRL with static workflow baselines across multiple Text-to-SQL benchmarks. Overall, SquRL consistently outperforms all static baselines across different parameter budgets, demonstrating the effectiveness of dynamic workflow construction. Notably, even at small model scales, SquRL exhibits strong gains: SquRL-1.5B already surpasses the best static workflow on both Spider-Dev and Spider-Test, indicating that dynamic workflow selection can compensate for limited model capacity. As the model size increases, this advantage becomes more pronounced, suggesting that larger models are better able to exploit the flexibility offered by dynamic workflow construction. At the largest scale, SquRL-7B achieves the best overall performance across all benchmarks, reaching 86.27% on Spider-Dev, 86.58% on Spider-Test, and 67.60% on Bird-Dev. Taken together, these results demonstrate that dynamic workflow construction provides consistent and scalable improvements, with pronounced improvements on challenging queries. Additionally, as results shown in Table 2, SquRL-7B achieves the highest 44.97% score on the challenging Spider2.0-Lite benchmark. To en-

[2]https://huggingface.co/Qwen/Qwen2.5-7B-Instruct

*Table 2.* Performance Comparison of Different Methods on Spider2.0-Lite benchmark.

| Methods | Accuracy (%) |
|---|---|
| DIN-SQL + GPT-4o | 1.46 |
| CHESS + GPT-4o | 3.83 |
| DailSQL + GPT-4o | 5.68 |
| LinkAlign + Deepseek-R1 | 33.09 |
| RSL-SQL + Deepseek-R1 | 33.09 |
| SquRL-7B + Deepseek-R1 | **49.18** |

sure a fair comparison with prior methods, we align the actor backbone by adopting DeepSeek-R1 as the API model, and the resulting performance highlights SquRL's effectiveness in handling complex, real-world queries.

> **Main finding for Q2**
>
> *SquRL learns a robust policy that dynamically selects effective workflows across diverse Text-to-SQL benchmarks, consistently identifying near-optimal workflow choices under varying query distributions.*

## 5.3. Further Analysis

**Part I: Complexity Analysis.** We evaluate SquRL across difficulty levels using the SynSQL dataset which is partitioned into Simple, Moderate, Complex, and Highly Complex subsets. As shown in Table 1, SquRL consistently outperforms static baselines at all difficulty levels, with performance gap scaling with query complexity. While the dynamic policy achieves a modest +4.05% improvement on Simple queries (82.06% *vs.* 78.01%), the performance gap increases to +7.97% on Highly Complex queries where SquRL-7B achieves 46.51% accuracy compared to RSL-SQL's 38.54%. These results show that dynamic workflows are more effective and generalized on complex queries that require diverse reasoning strategies which cannot be handled by single static workflows.

*Table 3.* Workflow selection behavior across query types on SynSQL. Accuracy is execution accuracy and latency is measured in seconds.

| Method | Query Type | Avg. Actors/WF | Avg. Length/WF | Accuracy | Latency |
|--------|-----------|----------------|----------------|----------|---------|
| Qwen-7B | Simple | 4.24 | 1.41 | 58.26 | 29.56 |
| Qwen-7B | Complex | 5.37 | 1.54 | 23.28 | 28.87 |
| SquRL-7B | Simple | 1.16 | 1.16 | 82.06 | 10.71 |
| SquRL-7B | Complex | 2.74 | 1.61 | 59.47 | 16.34 |

**Part II: Backend Analysis.** The frozen backend model that executes the dynamically generated workflows directly determines reward quality during SquRL training, thereby influencing policy optimization. As demonstrated in Table 4, stronger backends consistently yield superior performance across all benchmarks. These results suggest that weaker backends are more likely to generate invalid SQL during complex workflow execution, which reduces reward accuracy and hinders effective learning. Moreover, SquRL derives substantial benefits from stronger backends that provide more reliable execution feedback, particularly for complex and long-horizon query distributions.

*Table 4.* Impact of backend models on SquRL performance. We use Qwen2.5-1.5B-Instruct as the policy model with different backend models executing the generated workflows during training.

| API Backbone | Spider-Dev | Spider-Test | Bird-dev |
|--------------|-----------|-------------|----------|
| Qwen-Plus | 85.10 | 84.83 | 64.47 |
| Qwen-Turbo | 83.25 | 83.97 | 61.69 |
| GLM-4.7 | 83.56 | 82.78 | 63.14 |
| Deepseek-V3.2 | 85.80 | 85.95 | 63.10 |

**Part III: Dynamic Masking Analysis.** We evaluate the robustness of SquRL under varying actor availability by adjusting the retention rate r from 0.1 to 1.0, where r=1.0 indicates that all actors are accessible for workflow construction. Figure 3 compares the performance of models trained with and without dynamic actor masking across different retention rates. SquRL models trained with dynamic masking exhibit robust performance across all retention rates, while those trained on the full actor pool degrade sharply when fewer actors are available. These results indicate that dynamic masking prevents overfitting to specific high-performing actors, enabling the policy to learn flexible compensatory strategies.

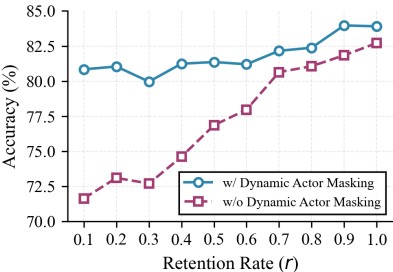

*Figure 3.* Performance comparison with and without Dynamic Actor Masking (DAM) across different retention rates.

**Part IV: Pseudo Reward Analysis.** We investigate the impact of pseudo rewards at different $p$, which is the probability of replacing the real execution reward with LLM-evaluated scores. As shown in Table 5, using a relatively small replacement ratio such as $p = 0.1$ unexpectedly improves performance across both benchmarks compared to relying solely on true rewards. This occurs because pseudo rewards provide denser feedback signals, enabling the policy to learn from partially correct workflows that would otherwise yield no learning signal. However, when $p$ increases to 0.3, performance drops substantially due to the accumulation of noisy rewards that misguide policy learning.

*Table 5.* Performance comparison with different $p$ ratios.

| Methods | Spider-Dev | Bird-Dev |
|---------|-----------|----------|
| True Only | 82.71 | 63.61 |
| 10% Pseudo | 85.10 | 64.47 |
| 30% Pseudo | 80.30 | 61.50 |

**Part V: Workflow Selection Analysis.** As shown in Table 3, we analyze how SquRL adapts workflow complexity across queries with different difficulty levels. Before RL training, the base model tends to construct unnecessarily complex workflows regardless of query difficulty, leading to higher computational cost without consistent performance gains. After RL training, SquRL exhibits clear difficulty-aware behavior: it favors compact workflows for simple queries while selecting more expressive workflows for complex queries. These results suggest that SquRL effectively matches workflow complexity to query difficulty, thus improving both efficiency and execution accuracy.

### 5.4. Ablation Study

We conduct an ablation study to analyze the individual contributions of supervised fine-tuning (SFT) and reinforcement learning (RL) in SquRL. As shown in Table 6, removing SFT reduces execution accuracy and increases format errors, confirming that SFT is essential for learning executable workflow structures. Removing RL further degrades accuracy and significantly increases runtime, indicating that RL is critical for refining workflows into efficient solutions. When both components are removed, the model degenerates into a basic instruction-follower incapable of generating valid workflows. These results demonstrate that SquRL's performance relies on the synergy between SFT and RL

rather than either component alone.

*Table 6.* Ablation study isolating the contributions of SFT and RL in SquRL. Removing either component degrades performance, with the combined removal reducing the system to a generic instruction-following model.

| Method | Acc.(%) | Format Err.(%) | Time(s) |
|--------|---------|----------------|---------|
| All | 83.27 | 0.29 | 4.41 |
| w/o SFT | 78.94 | 1.26 | 5.26 |
| w/o RL | 77.37 | 3.87 | 10.51 |
| w/o both | 18.47 | 77.37 | 7.19 |

# 6. Conclusion

We present SquRL, a reinforcement learning framework that enables dynamic workflow construction for Text-to-SQL. Through theoretical analysis and empirical validation, we demonstrate that adaptive workflow selection consistently outperforms any static pipeline. Experiments show that SquRL achieves substantial improvements on complex and out-of-distribution queries, establishing dynamic policy as a principled alternative to manual workflow design.

## Impact Statement

This paper presents work whose goal is to advance the field of machine learning. There are many potential societal consequences of our work, none of which we feel must be specifically highlighted here.

## Acknowledgments

This work was partially supported by the National Natural Science Foundation of China under Grant No. 62506077, 72201061, and 72271233; Suzhou Key Laboratory of Artificial Intelligence and Social Governance Technologies (SZS2023007), Smart Social Governance Technology and Innovative Application Platform (YZCXPT2023101), and the Innovation System of the Integration between Industry and Education for Smart Governance (CJRH2024101).

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

# A. Related Work

**Text-to-SQL Systems and Methodologies**  Text-to-SQL has evolved from rule-based systems (Li & Jagadish, 2014) to deep learning approaches (Sutskever et al., 2014; Vaswani et al., 2017; Zhong et al., 2017), and recently to large language model-based methods. Early neural architectures incorporated intermediate representations (Choi et al., 2021) and graph structures (Bogin et al., 2019) for structural reasoning. Pre-trained language models such as BERT (Devlin et al., 2019) and specialized variants (Yin et al., 2020; Scholak et al., 2021) enabled schema-aware encoding (Li et al., 2023a;b), establishing strong baselines. The emergence of LLMs introduced two primary paradigms: in-context learning (ICL) and fine-tuning. Representative ICL methods include DIN-SQL (Pourreza & Rafiei, 2023a), which decomposes queries into multi-stage pipelines with self-correction; DAIL-SQL (Gao et al., 2024), which optimizes few-shot selection via question skeleton matching; C3 (Dong et al., 2023), combining schema linking with calibration bias prompting; and MAC-SQL (Wang et al., 2025a), orchestrating multi-agent collaboration. Recent advances introduced enhanced schema linking (Cao et al., 2024; Caferoglu & Ulusoy, 2024; Wang et al., 2025c), chain-of-thought reasoning (Zhang et al., 2023; Tai et al., 2023), and self-consistency mechanisms (Ren et al., 2024; Wang et al., 2022). Fine-tuning approaches adapt open-source models through specialized pre-training (Li et al., 2024), multi-task decomposition (Pourreza & Rafiei, 2024), or domain-specific data augmentation (Chen et al., 2024b). However, all existing methods—whether manually engineered or learned—adopt fixed, static workflows that remain unchanged across queries at inference time, fundamentally limiting adaptability to heterogeneous distributions and out-of-distribution scenarios (Pourreza & Rafiei, 2023b).

**Reinforcement Learning for Code Generation**  Reinforcement learning has proven effective for improving code generation through execution feedback (Le et al., 2022; Chen et al., 2021). In text-to-SQL, RL methods leverage execution results (Zhong et al., 2017), train verifiers from execution traces (Ni et al., 2023), or enable self-debugging via error explanations (Chen et al., 2024a). Execution-guided approaches employ minimum Bayes risk decoding (Shi et al., 2022) and iterative refinement (Pourreza & Rafiei, 2023a) to select high-quality candidates. However, existing RL-based text-to-SQL methods optimize token-level policies $\pi_{\text{SQL}}(y \mid q)$ that directly generate SQL statements (Ma et al., 2025; Yao et al., 2025; Pourreza et al., 2025), tightly coupling reasoning and generation into monolithic models. This formulation precludes adaptive strategy selection—for instance, routing simple queries through lightweight pipelines while deploying multi-agent decomposition for complex nested queries. In contrast, our work learns a policy over structured workflow space rather than token space, separating **how to reason** from **what to generate**, thereby enabling dynamic orchestration of modular components as reusable building blocks.

# B. Theoretical upper bound of dynamic workflow orchestration

### B.1. Problem setup

Let queries $q$ be drawn i.i.d. from an unknown distribution $\mathcal{D}$ over $\mathcal{Q}$. Let $\mathcal{W} = \{W_1, \ldots, W_K\}$ be a finite set of workflows. For each workflow $W_i$ define the Bernoulli random variable

$$Y_i(q) := \mathbb{I}\{W_i(q) \text{ executes correctly}\} \in \{0, 1\}.$$

Denote the success region of $W_i$ by $\mathcal{A}_i := \{q \in \mathcal{Q} : Y_i(q) = 1\}$. Let

$$p_i := \mathbb{E}_{q \sim \mathcal{D}}[Y_i(q)] = \Pr(q \in \mathcal{A}_i).$$

The best static workflow's expected execution accuracy is

$$\text{EX}_{\text{static}} := \max_{1 \le i \le K} p_i.$$

The oracle dynamic selector (which for each query chooses any workflow that succeeds when possible) achieves

$$\text{EX}_{\text{dyn}} := \mathbb{E}_{q \sim \mathcal{D}}\Big[ \max_{1 \le i \le K} Y_i(q) \Big] = \Pr\Big( \bigcup_{i=1}^{K} \mathcal{A}_i \Big).$$

Define the gap

$$\Delta := \text{EX}_{\text{dyn}} - \text{EX}_{\text{static}}.$$

## B.2. Main theorem and proof

**Theorem B.1** (Non-negativity and characterization of equality). *For any finite workflow set $\mathcal{W}$, $\Delta \geq 0$. Moreover, $\Delta = 0$ if and only if there exists an index $i^*$ such that $\Pr\left(\bigcup_{i=1}^{K} \mathcal{A}_i \setminus \mathcal{A}_{i^*}\right) = 0$; equivalently, the success region of some single workflow covers the union of all success regions almost surely.*

*Proof.* For any fixed $q$ and any $i$ we have $\max_j Y_j(q) \geq Y_i(q)$. Taking expectation and then maximizing over $i$ gives

$$\mathrm{EX}_{\mathrm{dyn}} = \mathbb{E}[\max_j Y_j(q)] \geq \max_i \mathbb{E}[Y_i(q)] = \mathrm{EX}_{\mathrm{static}},$$

hence $\Delta \geq 0$.

Let $i^*$ be an index achieving $\mathrm{EX}_{\mathrm{static}} = \Pr(\mathcal{A}_{i^*})$. By definitions,

$$\mathrm{EX}_{\mathrm{dyn}} = \Pr\left(\bigcup_{i=1}^{K} \mathcal{A}_i\right).$$

Thus

$$\Delta = \Pr\left(\bigcup_{i=1}^{K} \mathcal{A}_i\right) - \Pr(\mathcal{A}_{i^*}) = \Pr\left(\bigcup_{i=1}^{K} \mathcal{A}_i \setminus \mathcal{A}_{i^*}\right).$$

Therefore $\Delta = 0$ exactly when $\Pr\left(\bigcup_{i=1}^{K} \mathcal{A}_i \setminus \mathcal{A}_{i^*}\right) = 0$, i.e., $\mathcal{A}_{i^*}$ covers the union almost surely. This proves the characterization. $\square$

## B.3. Quantitative relations with pairwise disagreement

Define pairwise disagreement probabilities

$$D_{ij} := \Pr_{q \sim \mathcal{D}}(Y_i(q) \neq Y_j(q)) = \Pr(\mathcal{A}_i \triangle \mathcal{A}_j),$$

where $\triangle$ denotes symmetric difference.

Fix $i^* = \arg\max_i p_i$ (choose any maximizer). For any $j$ we have the exact identity

$$\Pr(\mathcal{A}_j \setminus \mathcal{A}_{i^*}) = \frac{p_j - p_{i^*} + D_{i^* j}}{2}.$$

Consequently,

$$\Delta = \Pr\left(\bigcup_{i=1}^{K} \mathcal{A}_i \setminus \mathcal{A}_{i^*}\right) \geq \max_j \Pr(\mathcal{A}_j \setminus \mathcal{A}_{i^*}) = \max_j \frac{p_j - p_{i^*} + D_{i^* j}}{2}.$$

*Remark.* The displayed identity follows by writing $D_{i^* j} = \Pr(\mathcal{A}_{i^*} \setminus \mathcal{A}_j) + \Pr(\mathcal{A}_j \setminus \mathcal{A}_{i^*})$ and using $p_{i^*} = \Pr(\mathcal{A}_{i^*})$, $p_j = \Pr(\mathcal{A}_j)$.

The inequality shows that disagreement with the best static workflow (as measured by $D_{i^* j}$) contributes positively to the gap; however, it also highlights that *pairwise disagreement alone is not sufficient* to guarantee a positive gap unless that disagreement yields mass outside the best static's success region. In particular, it is possible to have $D_{ij} > 0$ for some $i, j$ while $\Delta = 0$ (see below).

## B.4. Counterexample (pairwise disagreement but zero gap)

Let $\mathcal{Q} = \{a, b\}$ with $\Pr(a) = \Pr(b) = 1/2$. Define workflows with success sets $\mathcal{A}_1 = \{a\}$, $\mathcal{A}_2 = \{b\}$, and $\mathcal{A}_3 = \{a, b\}$. Then $D_{12} = 1 > 0$ but $\bigcup_i \mathcal{A}_i = \mathcal{A}_3$ and hence $\Delta = 0$ because workflow 3 (the static best) already covers all solvable queries. This example demonstrates that pairwise disagreement by itself does not imply $\Delta > 0$.

### B.5. Discussion

The exact expression $\Delta = \Pr(\bigcup_i \mathcal{A}_i) - \max_i \Pr(\mathcal{A}_i)$ gives the conceptual interpretation: the oracle dynamic selector attains the measure of the *union* of success regions, while any single static workflow can only attain the measure of one success region. Thus the dynamic advantage arises precisely from queries that are solvable by *some* workflow but not by the single best static workflow. The quantitative identities above allow empirical estimation of how much of the union lies outside the best static success set, and also relate this to pairwise disagreement $D_{ij}$ when useful.

## C. Experimental Validation of the Theoretical Upper Bound

### C.1. Workflow Set and Template Diversity

To empirically validate the theoretical upper bound of dynamic workflow orchestration, we construct a diverse workflow set that spans a wide range of reasoning strategies, computational budgets, and control structures. Each workflow is instantiated from a high-level *template*, which specifies the abstract composition and execution order of functional actors. By analyzing templates rather than concrete actor instantiations, we expose the fundamental structural differences among workflows, enabling a principled study of workflow heterogeneity.

**Actor Primitives.** We begin by briefly introducing the functional roles of each actor primitive used across templates:

- **Reduce actor.** Eliminates redundant schema elements from large-scale databases to ensure that downstream reasoning operates over a compact and relevant schema subset, improving both efficiency and focus.

- **Parse actor.** Performs schema linking by identifying relevant tables and columns for the given query, providing attention signals that guide subsequent SQL generation.

- **Generate actor.** Produces complete SQL statements directly from the query and schema, encapsulating standard end-to-end Text-to-SQL generation methods.

- **Decompose actor.** Breaks complex queries into a sequence of logically progressive sub-questions, enabling compositional reasoning and interpretable multi-step problem solving.

- **Scale actor.** Generates diverse, high-quality SQL candidates via multi-sample decoding or parallel generation, increasing the probability of covering the gold SQL through broader exploration.

- **Optimize actor.** Refines candidate SQL queries using environmental feedback (e.g., execution errors or result mismatches), iteratively correcting syntax or semantic inconsistencies.

- **Select actor.** Chooses the optimal SQL statement from a candidate pool, typically in collaboration with the scale actor, enabling convergence from broad exploration to a single high-precision output.

These actors span the full reasoning lifecycle, from schema grounding and generation to exploration, refinement, and final decision making.

**Template Set.** We design a set of templates that vary systematically along several dimensions: (i) whether schema grounding is explicit (parser), (ii) whether generation is single-shot or diversified (generator vs. scaler), (iii) whether feedback is exploited (optimizer), (iv) whether decision-making is centralized (selector), and (v) whether control flow is linear or branched. The full template set is shown below.

- **Template 0:** [generator]. A single-pass, end-to-end generator with no explicit schema grounding, diversification, or feedback.

- **Template A:** [generator, optimizer]. A linear pipeline that augments direct generation with execution-based refinement.

- **Template B:** [[generator, generator, generator], selector]. A parallel ensemble of generators followed by a selector, emphasizing diversity and empirical selection without feedback loops.

- **Template C:** $[\text{generator}, [\text{optimizer}, \text{optimizer}], \text{selector}]$. A hybrid structure that performs generation followed by multiple rounds of feedback-driven refinement before final selection.

- **Template D:** $[\text{parser}, \text{generator}]$. A schema-grounded pipeline where parsing precedes generation, without diversification or feedback.

- **Template E:** $[\text{parser}, [\text{scaler}, \text{scaler}], \text{optimizer}, \text{selector}]$. A schema-grounded, diversity-first workflow with multi-sample generation, followed by refinement and final selection.

- **Template F:** $[[\text{scaler}, \text{scaler}, \text{scaler}, \text{scaler}], \text{selector}]$. A large-scale exploration template emphasizing massive candidate generation and empirical selection, without explicit parsing or refinement.

- **Template G:** $[[\text{generator}, \text{generator}], [\text{scaler}, \text{scaler}], \text{selector}]$. A mixed ensemble that combines deterministic generation with diversified scaling before selection.

- **Template H:** $[\text{parser}, \text{generator}, [\text{scaler}, \text{scaler}, \text{scaler}], \text{optimizer}, \text{selector}]$. A deeply structured workflow that integrates schema grounding, initial generation, large-scale diversification, feedback-driven refinement, and final decision making.

- **Template I:** $[\text{parser}, [\text{generator}, \text{scaler}], [\text{optimizer}, \text{optimizer}], \text{selector}, \text{optimizer}]$. A multi-stage feedback-centric pipeline that interleaves generation, scaling, repeated optimization, and selection, reflecting maximal structural complexity.

### C.2. Empirical Estimation of Workflow Distances

To empirically validate the theoretical analysis in Section 3, we estimate the pairwise distances between the ten workflows introduced in Section C.1 using a finite sample of queries drawn from the SynSQL dataset. The effectiveness disagreement between workflows $W_i$ and $W_j$ denotes the probability of producing different correctness outcomes:

$$D_{\text{sample}}(i, j) = \Pr_{q \sim \mathcal{D}} (Y_i(q) \neq Y_j(q)) \tag{21}$$

Moreover, to capture efficiency differences between two workflows, we introduce the efficiency disagreement:

$$D_{\text{efficiency}}(i, j) = \mathbb{E}_{q \sim \mathcal{D}} \left[ \frac{|t_i(q) - t_j(q)|}{t_i(q) + t_j(q)} \right] \tag{22}$$

Where $t_i$ denotes the expected runtime of workflow $W_i$ on query $q$. Therefore, we can define the overall distance between two workflows as:

$$D(W_i, W_j) = \frac{1}{2} \left( D_{\text{sample}}(i, j) + D_{\text{efficiency}}(i, j) \right) \tag{23}$$

Since the true expectations in Equations (21)–(23) are intractable, we approximate them via sample averages.

For the effectiveness disagreement defined in Equation (21), we compute the unbiased empirical estimator:

$$\hat{D}_{\text{sample}}(i, j) = \frac{1}{N} \sum_{k=1}^{N} \mathbb{I}\{Y_i(q_k) \neq Y_j(q_k)\}. \tag{24}$$

Here, $\{q_k\}_{k=1}^{N}$ denotes the sampled query set and $Y_i(q_k)$ indicates whether workflow $i$ executes correctly on query $q_k$.

Similarly, for the efficiency disagreement defined in Equation (22), we estimate it using:

$$\hat{D}_{\text{efficiency}}(i, j) = \frac{1}{N} \sum_{k=1}^{N} \frac{|t_i(q_k) - t_j(q_k)|}{t_i(q_k) + t_j(q_k)}. \tag{25}$$

The overall workflow distance is then estimated as:

$$\hat{D}(i, j) = \frac{1}{2} \left( \hat{D}_{\text{sample}}(i, j) + \hat{D}_{\text{efficiency}}(i, j) \right), \tag{26}$$

consistent with Equation (23) in the main text. Then, we compute these distances for all pairs of workflows using the ten workflows defined in Section C.1 over the same evaluation corpus described in Section 3 The resulting pairwise distance matrix is visualized in Figure 4.

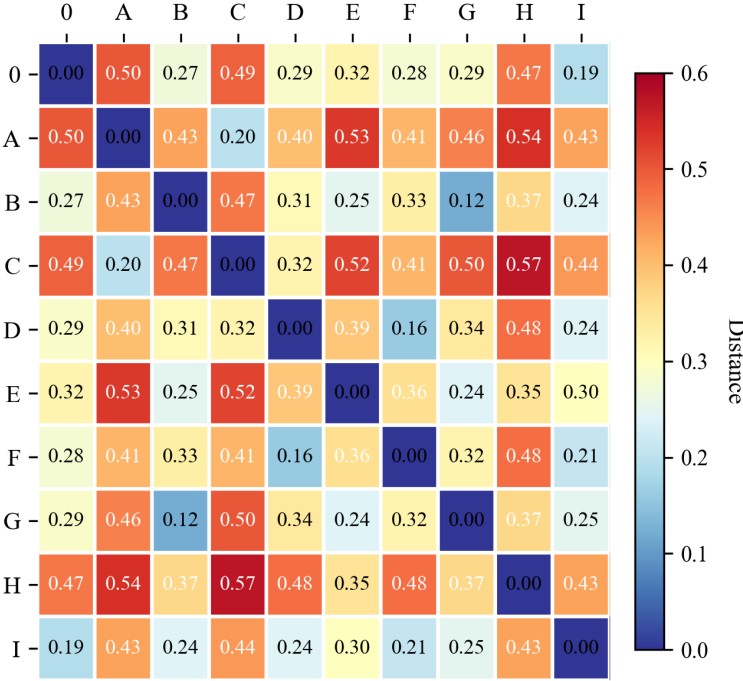

*Figure 4.* Pairwise distance matrix between different workflows.

### C.3. Empirical Upper Bound on Efficiency

In this subsection, we empirically estimate the upper bound of efficiency gains achievable through dynamic workflow selection, as motivated in Section 4.

**Experimental Design.** For each query $q$, we record the execution outcomes and runtimes of all $M = 10$ workflows. Based on correctness alone, we compute the number of workflows that successfully execute $q$, denoted by $N(q) \in \{1, \ldots, 10\}$. This value serves as a proxy for problem difficulty: $N = 1$ corresponds to extremely sparse cases where only a single workflow succeeds, while larger $N$ indicates easier queries solvable by many workflows.

We then group all queries according to their $N(q)$ values. For each difficulty level $N$, we compute:

- the *average maximum runtime*, corresponding to the case where the slowest correct workflow is always selected;

- the *average minimum runtime*, corresponding to the oracle case where the fastest correct workflow is always selected.

These two quantities define the empirical upper and lower bounds on runtime achievable through workflow selection under correctness constraints.

**Efficiency Upper Bound.** Let $\mathcal{Q}_N$ denote the set of queries for which exactly $N$ workflows execute correctly. For each $q \in \mathcal{Q}_N$, let $t_i(q)$ be the runtime of workflow $i$ and $\mathcal{C}(q)$ be the set of workflows that execute correctly on $q$. We define:

$$t_{\max}(q) = \max_{i \in \mathcal{C}(q)} t_i(q), \qquad t_{\min}(q) = \min_{i \in \mathcal{C}(q)} t_i(q). \tag{27}$$

We then compute the empirical averages:

$$\bar{t}_{\max}(N) = \frac{1}{|\mathcal{Q}_N|} \sum_{q \in \mathcal{Q}_N} t_{\max}(q), \qquad \bar{t}_{\min}(N) = \frac{1}{|\mathcal{Q}_N|} \sum_{q \in \mathcal{Q}_N} t_{\min}(q). \tag{28}$$

The efficiency upper bound at difficulty level $N$ is then defined as the relative improvement:

$$\Delta_{\text{eff}}(N) = \frac{\bar{t}_{\max}(N) - \bar{t}_{\min}(N)}{\bar{t}_{\max}(N)}. \tag{29}$$

**Results and Analysis.**  Figure 5 reports the empirical distributions of $|\mathcal{Q}_N|$ (right panel) as well as $\bar{t}_{\max}(N)$ and $\bar{t}_{\min}(N)$ (left panel) across all difficulty levels $N \in \{1, \ldots, 10\}$. We observe that:

- The majority of queries fall into low-to-moderate $N$ regimes, indicating that most queries are solvable by only a small subset of workflows.

- The gap between $\bar{t}_{\max}(N)$ and $\bar{t}_{\min}(N)$ is substantial for most $N$, yielding large efficiency upper bounds, often exceeding $70\%$.

- As $N$ decreases, i.e., queries become more difficult and sparse, the efficiency upper bound initially remains high but drops sharply when $N = 1$, reflecting the fact that no selection is possible when only a single workflow succeeds.

These results empirically confirm that dynamic workflow selection admits a significant efficiency upper bound, especially in regimes where multiple workflows are correct but exhibit heterogeneous runtime behaviors. This observation directly supports **Key Insight Two** in Section 3, demonstrating that dynamic orchestration can substantially outperform any fixed static workflow in efficiency while preserving correctness.

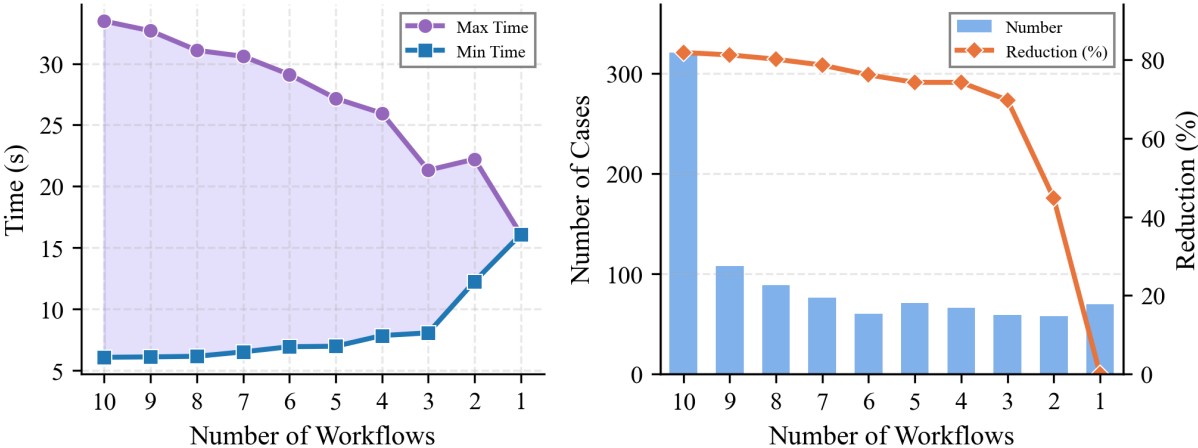

*Figure 5.* **Performance metrics across task difficulty levels.** (Left) Max/min execution times versus task difficulty (measured by number of capable workflows). (Right) Case distribution and time reduction by difficulty. Easier tasks (left) show greater time variation and more samples, while harder tasks (right) demonstrate convergent execution times.

## D. Supplementary Experimental Setup

### D.1. Dataset

**SPIDER**  is a cross-domain benchmark for complex Text-to-SQL parsing, consisting of 10,181 questions and 5,693 unique SQL queries over 200 multi-table databases spanning 138 domains. It is designed to evaluate cross-database generalization, with particular emphasis on compositional reasoning, multi-table joins, and nested queries.

**BIRD**  contains 12,751 question–SQL pairs across 95 large databases covering 37 professional domains. It introduces realistic challenges such as noisy data, value grounding, and external knowledge integration, substantially increasing the difficulty of schema linking and query generation compared to SPIDER.

**SPIDER 2.0**  comprises over 600 real-world Text-to-SQL tasks drawn from enterprise databases with thousands of columns and diverse SQL dialects. It extends beyond single-query parsing to multi-step workflows involving schema metadata, external documentation, and execution feedback, posing significantly greater challenges than both SPIDER and BIRD.

**SYNSQL-2.5M** is a large-scale synthetic Text-to-SQL dataset containing 2,544,390 question–SQL pairs across 16,583 databases spanning diverse real-world domains. Databases are automatically synthesized from web tables using large language models, with SQL queries generated at varying complexity levels and natural language questions produced in multiple linguistic styles. Notably, SYNSQL-2.5M is the first million-scale Text-to-SQL dataset to provide chain-of-thought solutions for each sample, enhancing training quality and model interpretability.

## E. Case Study

To qualitatively analyze how RL influence workflow construction, we review representative examples from SynSQL before and after training.

---

**Reasoning Example of the Untrained Qwen2.5-7B-Instruct for Complex Task.**

```
<think>
```
1. **Template Selection**: The query involves selecting specific columns from the `experiments` table and filtering based on certain conditions. It also requires ordering and limiting the results. Given the complexity and the need for multiple steps, Template I seems most suitable as it allows for a detailed pipeline with parsing, generation, scaling, optimization, and selection.

2. **Actor Selection**:

- **Parser**: We need to identify the relevant tables and columns. The `experiments` table is the primary focus, and we need to consider the `performance_metrics` table for evaluation metric values.

- **Generator**: We need to generate the SQL query based on the parsed schema links.

- **Scaler**: We need to generate diverse SQL candidates to increase the probability of covering the gold SQL.

- **Optimizer**: We need to refine the generated SQL based on execution feedback.

- **Selector**: We need to select the optimal SQL statement from the candidates.

3. **Pipeline Composition**:

- **Parser**: `MACSQLCoTParser` is a good choice for its chain-of-thought schema parsing.

- **Generator**: `MACSQLGenerator` is suitable for its multi-agent collaboration system.

- **Scaler**: `MACSQLScaler` is appropriate for its multi-agent SQL generation strategy.

- **Optimizer**: `MACSQLOptimizer` is effective for its iterative refinement using execution feedback.

- **Selector**: `CHESSSelector` is a good choice for its execution-based selection.

The final pipeline will be:

- Parse the schema to identify relevant tables and columns.

- Generate an initial SQL query.

- Scale the generated SQL to create diverse candidates.

- Optimize the candidates based on execution feedback.

- Select the best SQL statement.

```
</think>
<answer>list[["MACSQLCoTParser", "MACSQLGenerator", "MACSQLScaler",
"MACSQLOptimizer", "CHESSSelector"]]</answer>
```

---

---

**Reasoning Example of the Trained Qwen2.5-7B-Instruct for Complex Task.**

```
<think>
```
Upon examining the natural language question, it is evident that the query requires filtering experiments based on specific criteria: classification type, completion status, and top-10 ranking according to evaluation metric values. The output must include multiple attributes such as experiment name, evaluation metric values, embedding dimensions, statuses, creators, and last modifiers. This indicates that the query involves complex filtering, ranking, and projection across multiple tables, particularly the `experiments` table which contains both the evaluation metrics and metadata.

The schema reveals that the `experiments` table is central to this query, containing columns like `experiment_name`, `value` (likely the evaluation metric), `embedding_dim`, `state`, `created_by`, `last_modified_by`, and `last_modified_date`. Additionally, the `performance_metrics` table may be necessary for retrieving specific evaluation metric values, though it is not directly referenced in the question. However, since the `value` column in `experiments` is mentioned and assumed to represent the evaluation metric, direct access to `performance_metrics` is not strictly required.

Given the complexity of the query—specifically the need to rank experiments by evaluation metric value and select the top 10—it is clear that a single generator would struggle to produce the correct SQL without proper schema linking and candidate diversification. The presence of multiple relevant columns across different tables necessitates precise schema linking to avoid ambiguity, especially when dealing with foreign keys and relationships.

Now, considering the available templates, Template A (`[generator]`) would be insufficient because it lacks mechanisms for schema linking or candidate diversification, which are critical for handling complex queries with multiple join paths and ranking conditions. Similarly, templates like Template C, F, G, H, I, and J introduce unnecessary parallelism or multiple optimizers/selectors, which could lead to inefficiency or redundancy given the straightforward nature of the SQL generation task. Template B (`[generator, optimizer]`) introduces an optimizer, which would allow refinement based on execution feedback. However, without an initial parser to identify relevant schema elements, the optimizer might operate on incomplete or incorrect schema links, leading to suboptimal corrections. Therefore, while an optimizer can improve robustness, it requires a reliable generator as its input. Template E (`[parser, generator]`) includes a parser to extract relevant schema links before SQL generation. This is necessary because the query involves multiple tables and columns with indirect relationships (e.g., `experiments` referencing `performance_metrics` through `value`). A parser like `RSLSQLBiDirParser` or `CHESSSelectorParser` would help identify that only the `experiments` table is needed for the core logic, reducing noise from irrelevant tables.

Given these considerations, the optimal configuration must include both a parser for schema linking and a generator capable of producing high-quality SQL with minimal errors. Among the available generators, `MACSQLGenerator` supports multi-agent collaboration and iterative refinement, making it robust even without an optimizer if used in conjunction with a strong parser.

Therefore, the solution must consist of a parser followed by a generator. The parser should be capable of adaptive schema linking, and the generator should handle complex filtering and ranking. The final pipeline must end with a generator, as the output must be a single SQL statement, not a set of candidates.

Based on this reasoning, the correct template is Template E: `[parser, generator]`. The parser must be a strong, adaptive schema linker, and the generator must be a multi-agent system that can handle complex logic and produce accurate SQL. Among the available actors, `LinkAlignParser` provides adaptive configuration and external knowledge integration, while `MACSQLGenerator` supports iterative refinement and schema pruning.

Thus, the final pipeline must be: `[LinkAlignParser, MACSQLGenerator]`.
```
</think>
<answer>list['LinkAlignParser', 'MACSQLGenerator']</answer>
```

## F. Computational Cost

As shown in Table 5, the primary training overhead of SquRL arises from reward evaluation, since each RL rollout requires executing the generated workflow and assessing its execution outcome. Compared with conventional token-level RL methods, this introduces additional environment interaction cost because workflow execution may involve multiple actor calls and intermediate SQL refinement stages. Nevertheless, the overall training cost remains practically manageable due to

the relatively small policy model and the parallelizable nature of workflow execution.

During inference, the computational cost scales with the number of sampled workflows, while latency increases sublinearly because workflows can be executed concurrently. For example, increasing the number of sampled workflows from one to five leads to approximately $5.2\times$ higher token consumption but only $1.98\times$ higher latency. This behavior indicates that dynamic workflow construction introduces favorable scalability properties under parallel execution environments.

| Setting | GPU Hours | API Tokens | Latency |
|---|---|---|---|
| RL Training | 0.124 | 502K | 190.22s |
| Inference (1 WF) | – | 8,038 | 10.51s |
| Inference (5 WF) | – | 41,783 | 20.82s |

*Table 7.* Training and inference cost of SquRL. Training cost is reported per RL step, and inference cost is reported per query. "WF" denotes workflow. Inference GPU hours are omitted because it introduces negligible overhead relative to training stage.

# G. LLM-based Pseudo Reward Design

SquRL introduces LLM-based pseudo rewards to reduce the cost of reward evaluation during RL training. Since real rewards require full workflow execution, rollout evaluation is computationally expensive and slows policy optimization. To mitigate this overhead, execution-based rewards are replaced with pseudo rewards with probability $p$. For each substituted rollout, the judge conducts a principle-guided pairwise comparison between the generated workflow and a baseline workflow under the same input. The principles are distilled from labeled historical trajectories and capture execution-relevant constraints, while the baseline serves as a reference for comparison. The judge's preference and confidence are then mapped to the pseudo reward described in Section 4.2.3. Because most policy updates still rely on real execution feedback, the noise introduced by pseudo rewards remains limited when $p$ is small. As shown in Table 5, pseudo rewards improve performance under a modest substitution ratio but become detrimental when the ratio is too large.

## G.1. Evaluation Principles

A direct LLM judge may rely on template-level plausibility while overlooking execution-critical constraints. In workflow-level Text-to-SQL, such constraints are essential: a workflow may omit SQL generation, leave parallel SQL candidates unresolved, or pass parser outputs to actors that require `pred_sql`. To bridge the gap between surface plausibility and actual execution behavior, we derive evaluation principles from labeled historical trajectories and provide them to the judge as an execution-aware rubric.

The principles are obtained through iterative self-evolution. In each iteration, the judge evaluates workflow pairs using the current principles. The resulting judgments are then compared with execution outcomes, and false-positive and false-negative cases are analyzed to identify recurring structural error patterns for principle refinement. After multiple iterations, we select the principles with the highest score. This process distills execution history into compact evaluation criteria rather than relying on an unconstrained prompt.

Table 8 quantifies the impact of the evolved principles. Simply scaling the judge model yields limited improvement, suggesting that model capacity alone is not the primary bottleneck. In contrast, introducing principles substantially improves the agreement between judge preferences and execution outcomes. These results indicate that execution-aware criteria help the judge distinguish workflows that are merely structurally plausible from those that are genuinely executable.

| Method | Success Rate (%) |
|---|---|
| Qwen-32B (small) | 50.4 |
| Qwen-235B (big) | 50.7 |
| Qwen-235B + baseline | 49.3 |
| Qwen-235B + principles | 56.9 |
| Qwen-235B + both | 57.0 |

*Table 8.* Agreement between LLM-judge preferences and real execution outcomes. Success rate measures whether the workflow preferred by the judge is also better under execution-based evaluation.

The final evolved principles used by the judge are summarized below.

---

**Evolved principles used by the LLM judge**

**1. Generator presence (hard rule).** Every executable workflow must contain at least one Generate actor that produces `pred_sql`, unless valid pre-existing SQL candidates are explicitly supplied and a Select actor consumes them. Workflows without a Generate actor are judged as failures unless this external grounding and type flow are both clear.

**2. Parallel branch closure (hard rule).** Parallel branches that produce `pred_sql` must be followed by a Select actor that resolves them into a single output. A parallel list of SQL-producing actors without a downstream selector is treated as an invalid workflow because execution has no well-defined terminal SQL.

**3. Type-flow integrity (hard rule).** Each actor must receive the input type it expects. Actors that require `pred_sql`, such as Optimizer or Selector, must consume outputs from Generate or Scale actors rather than parser or schema outputs. Demonstrable type mismatches are judged as execution failures.

**4. Generator-first signal.** Workflows are more reliable when the first SQL-producing actor is a Generate actor and Scale or Optimizer actors appear only after generation. This pattern avoids scaling or refining malformed non-SQL intermediate states and is a strong positive signal, especially for complex queries.

**5. Parser-generator coherence.** When a parser is used, its output should feed a subsequent Generate actor. Parser-only or parser-to-scale patterns are weak unless valid SQL candidates are already available. Parser-to-generator patterns are positive signals, particularly on large schemas.

**6. Baseline-informed simplicity.** Short workflows can be correct when they match the baseline structure and task context. For small databases or simple queries, absence of parser or optimizer should not be penalized unless the baseline indicates that these components are necessary.

**7. Positive structural signals.** The judge increases confidence for execution-relevant structures such as parsing before generation on large schemas, optimizer-selector chains after generation, and workflows that match the abstract shape of the reference baseline.

**8. External grounding for scale-only workflows.** Scale-only workflows without external `pred_sql` grounding are likely to fail, even if they contain optimizer or selector components. Without a Generate actor or explicit SQL candidates, scaling and selection operate on invalid inputs.

---

### G.2. Baseline Workflow

The baseline workflow serves as the reference point for pairwise judgment. Without such a reference, the judge must assess workflow quality in absolute terms, which is unstable across queries with different schemas and difficulty levels. Pairwise comparison instead evaluates whether the rollout workflow improves over a concrete alternative under the same query, schema, and actor context. When a verified workflow is available, we use it directly as the baseline; otherwise, a stronger LLM is used to generate a reference workflow. This stronger-LLM workflow is not used as the final Text-to-SQL solution, but solely as a training-time comparison signal.

Table 9 shows that the baseline workflow improves downstream policy optimization, despite not independently increasing judge-execution agreement in Table 8. This suggests that the baseline plays a role distinct from that of the principles. Rather than intrinsically improving judge accuracy, it changes which rollouts receive positive pseudo rewards by providing a stable comparison target. In particular, rollouts preferred over a competent baseline are more likely to reflect meaningful workflow structure, yielding a more informative exploration signal than random noise or baseline-free judgments.

| Pseudo Reward Source | Accuracy |
|---|---|
| Real Reward | 51.4 |
| Random Noise (10%) | 50.5 |
| Qwen3-235B w/o baseline (10%) | 52.1 |
| Qwen3-235B w/ baseline (10%) | 52.9 |

*Table 9.* Effect of the baseline workflow on downstream policy optimization. Accuracy is measured after training with the corresponding reward source.

# H. Prompt Template

## H.1. Dynamic Workflow Construction

We use the prompt template as below to instruct the LLM to generate a structured workflow based on the currect natural language query, database schema and available actors.

---

**System Prompt Configuration for Dynamic Workflow Construction**

`<|im_start|>system`
You are a strategic SQL Planning Agent. Your task is to analyze natural language queries and design an optimal Actor pipeline that produces correct SQL statements.
**# Available Actors:**
Below is the candidate `Actor` Pool available for this round. Do not select any `Actors` outside this list.
`{actors}`
**# Candidate Templates:**
Below are the candidate templates, which serve as slots to be filled with the selected `Actors`. You are encouraged to select templates from the candidate set; however, you may also use alternatives if they better suit task needs.
`{templates}`
**# Analysis Workflow**

1. **Template Selection**:

   - Analyze the natural language query and the database schema (complexity, table relationships, question type).
   - Determine the most suitable template(s) from the candidate set that can structure the Actor pipeline effectively.

2. **Actor Selection**:

   - Based on the selected template and the query characteristics, choose the Actors from the available pool that are best suited to handle each step of the task.
   - Consider the specific roles and capabilities of each Actor relative to the query.

3. **Pipeline Composition**:

   - Fill the selected template with the chosen Actors, arranging them sequentially or in parallel as required.
   - Ensure that the final Actor is `pred_sql` to produce the SQL output.

**# Output Requirements**

1. **Reasoning and Format**:

   - First, reason step by step to determine the final Actor list.
   - Provide your reasoning within `<think>...</think>`.
   - Provide the final result strictly within `<answer>...</answer>`.
   - The final answer must be a **Python list string**, enclosed exactly as `list[...]` inside `<answer>`.

2. **Actor Legality**:

   - Only use Actors from the `Available Actors`; any unlisted Actor is invalid.
   - The final pipeline must output `pred_sql` as the last Actor.

`<|im_end|>`

---

