# OpenReview forum: "Beyond Static Pipelines: Learning Dynamic Workflows for Text-to-SQL"
_ICML.cc/2026/Conference — ICML 2026 regular_

### Official Review · Reviewer_HWZH · 2026-03-10

**Soundness:** 3
**Presentation:** 3
**Significance:** 3
**Originality:** 3
**Overall Recommendation:** 4
**Confidence:** 3

**Summary:**

This paper proposes SquRL, a framework for dynamic workflow construction in Text-to-SQL. Instead of relying on a single fixed pipeline, the method represents Text-to-SQL systems as modular actors and templates, and trains an LLM to select a query-specific workflow conditioned on the input question and schema. The training pipeline consists of supervised fine-tuning on executable workflow labels followed by reinforcement learning with a shaped reward and two additional mechanisms, dynamic actor masking and pseudo rewards. The paper also gives a theoretical argument that oracle dynamic selection can outperform the best static workflow when candidate workflows have complementary success regions, and empirically reports gains over strong static baselines on Spider, BIRD, SynSQL, and Spider2.0-Lite.

**Compliance With Llm Reviewing Policy:**

Affirmed.

**Key Questions For Authors:**

> 1. How much of the learned routing policy transfers when the backend model or actor pool changes after training? For example, if a subset of actors is replaced by stronger implementations, or the backend LLM is updated, how much performance is retained without retraining? A convincing answer here would directly address W1.
>
> 2. Can the authors compare the current first-feasible SFT labeling strategy against stronger alternatives, such as labeling with the best successful workflow or supervising on multiple successful workflows per query? This would help clarify whether the current results depend on a biased initialization procedure, which is my concern in W2.
>
> 3. Can the authors provide stronger evidence that the shaped reward is aligned with the true target metric? In particular, I would like to see whether preferences induced by the intermediate rewards and pseudo rewards correlate well with held-out execution accuracy, since this is central to W3.
>
> 4. For the pseudo reward mechanism, what is the agreement rate between the judge model's pairwise preference and the actual execution outcome? If this agreement is weak or unstable across tasks, it would substantially affect my confidence in the RL objective.

**Limitations:**

No. The paper does not adequately discuss its most important technical limitations. In particular, it should more explicitly acknowledge the dependence of the learned policy on a fixed backend/tool stack, the bias introduced by first-feasible SFT labels, and the possibility that reward shaping and pseudo rewards optimize proxies that are not perfectly aligned with deployment-time execution accuracy.

**Strengths And Weaknesses:**

> Strengths:
>
> (S1) The problem is important and timely. The paper correctly identifies a real limitation of current Text-to-SQL systems: even strong methods are usually deployed as fixed pipelines, which is a poor match for heterogeneous query difficulty and real-world long-tail cases.
>
> (S2) The actor/template/workflow abstraction is a useful way to formalize the problem. It gives a clean modular view of existing systems and makes the dynamic routing problem concrete rather than purely conceptual.
>
> (S3) The paper combines theory and experiments in a reasonably compelling way. The theoretical analysis motivates why dynamic selection can outperform the best static workflow under complementary success regions, and the empirical results on Spider, BIRD, SynSQL, and Spider2.0-Lite suggest that the direction is promising, especially on more complex queries.
>
> (S4) The method itself is fairly well thought out from an engineering perspective. The combination of SFT initialization, RL with shaped execution feedback, dynamic actor masking, and pseudo rewards is coherent, and the paper includes useful ablations and sensitivity analyses.
>
> Weaknesses:
>
> (W1) My main concern is that the learned policy may be highly system-specific rather than genuinely learning transferable dynamic workflow selection. The policy is trained against rewards induced by a fixed actor pool, fixed backend models, and a fixed execution environment. In that setting, it is entirely plausible that the model is learning which workflow choices happen to work well for the current tool stack rather than learning a robust notion of which workflow is intrinsically appropriate for a query. This concern is reinforced by the backend sensitivity analysis: if changing the backend materially changes performance, then it is unclear how much of the learned policy would survive actor or model updates. In practice, this suggests the policy may need retraining whenever the underlying tools change, which weakens the paper's broader claim of learning dynamic workflow capability.
>
> (W2) The SFT supervision is heuristically constructed and appears biased in a way that may materially affect the learned policy. For each query, the paper searches templates from simpler to more expressive ones and uses the first successful workflow as the training label. This is not the same as supervising on the best workflow, or even on a set of near-optimal workflows. Instead, the policy is initialized to imitate the authors' search order and simplicity bias. Since RL in such a large combinatorial space is strongly shaped by initialization, I am not convinced that the later RL stage can fully undo this bias. The current experiments do not sufficiently establish that the method is learning the right routing policy rather than refining a biased first-feasible heuristic.
>
> (W3) The reward design is a significant methodological weakness. The paper claims to learn a policy for selecting the most appropriate workflow, but the training objective is a mixture of several proxies: format validity, executability, timeout behavior, execution correctness, runtime preference, and, for some rollouts, an LLM-based pseudo reward. This opens the door to optimizing for workflows that are easy to validate, efficient under the sandbox, or preferred by the judge model, rather than workflows that are truly best for final task success. I am especially concerned about the pseudo reward mechanism, because it replaces ground-truth execution feedback with pairwise judgments from another LLM and therefore changes the effective optimization target. The paper shows that larger pseudo-reward ratios already hurt performance, which suggests the approximation is fragile. More importantly, I do not think the current paper demonstrates that these shaped rewards are well aligned with final deployment-time execution accuracy. Without such evidence, it remains ambiguous whether SquRL learns dynamic workflow reasoning or exploits the specific reward structure.

---

> ### Author Rebuttal · Authors · 2026-03-31
>
> We thank the reviewer for the thorough and insightful feedback. We provide clarifications and additional evidence to further substantiate the effectiveness of SquRL.
>
> **Response to W1 & Q1**
>
> We thank the reviewer for this insightful concern. To mitigate the risk of system-specific overfitting, we introduce Dynamic Actor Masking (DAM) (Sec. 4.2.2), which randomly masks a subset of actors at each rollout. This exposes the policy to continuously varying actor pools rather than a fixed toolset, preventing reliance on specific actors and encouraging reasoning over functional roles, thereby improving generalization.
>
> As shown in Fig. 3 (Sec. 5.3), policies trained with DAM maintain stable performance under reduced actor availability at inference, whereas those without masking degrade sharply. This indicates that DAM promotes robustness to environmental perturbations rather than dependence on specific actors. We acknowledge that full transfer to entirely new toolchains is not guaranteed and may require adaptation; we will clarify this scope and leave broader cross-environment generalization to future work.
>
> Regarding backend sensitivity, stronger backends improve absolute accuracy, but the relative advantage of dynamic workflow selection remains consistent across backends (Sec. 5.3 Table 3), suggesting SquRL learns workflow organization rather than backend-specific shortcuts.
>
> ---
>
> **Response to W2 & Q2**
>
> We appreciate the reviewer's concern about simplicity bias in SFT and we address the concern from two perspectives.
>
> *Labeling strategy.* The “first-feasible” criterion reflects an efficiency preference: among equally accurate workflows, simpler ones are strictly preferable. However, this ordering may underexpose complex workflows. To mitigate this, we adopt a stratified search: 60% of samples start from Easy templates, 30% from Moderate, and 10% from Complex, ensuring adequate coverage of high-complexity reasoning paths.
>
> *RL as correction.* We agree that SFT alone may introduce bias. The subsequent RL stage compensates via large-scale rollout sampling and execution-based rewards, enabling exploration beyond the SFT distribution. In particular, “simple but failing” workflows are penalized on complex queries, driving the policy toward appropriate complexity.
>
> For Q2, supervising multiple valid workflows is a promising direction that we plan to explore in future work.
>
> ---
>
> **Response to Q3**
>
> We clarify that the multi-component reward is not a set of competing proxies but a hierarchical decomposition of the true objective: selecting workflows that are both correct and efficient. The reward follows a strict short-circuit structure: format validity → executability → correctness → efficiency. Each stage is evaluated only if prior conditions are satisfied, ensuring that invalid or non-executable workflows receive no correctness or efficiency credit. The efficiency term is applied only among correct workflows, preventing any preference for incorrect but efficient outputs.
>
> Thus, the reward is not an approximation but a direct operationalization of the task objective. This design is consistent with prior RL-based Text-to-SQL approaches (e.g., SQL-R1), which also employ execution-based structured rewards and show that intermediate signals improve training stability without altering the optimization target.
>
> To further address Q3, we are currently running ablation experiments to quantify the correlation between each reward component and held-out execution accuracy, and aim to share the full results before the end of the rebuttal period.
>
> >SQL-R1: Training Natural Language to SQL Reasoning Model By Reinforcement Learning
>
> ---
>
> **Response to Q4**
>
> We analyze the alignment between pseudo-rewards (LLM judge preferences) and execution outcomes (Table below). Specifically, we measure whether workflows preferred by the LLM judge are indeed better under real execution using "Success Rate".
>
> | Model                      | Success Rate (%) |
> |--------------------------|------------------|
> | Qwen-32B (small)         | 50.4             |
> | Qwen-235B (big)          | 50.7             |
> | Qwen-235B + baseline    | 49.3             |
> | Qwen-235B + principles   | 56.9             |
> | Qwen-235B + both         | 57.0             |
>
> The results show that `+both` and `+principles` both improve this consistency. Although `+baseline` reference alone does not noticeably improve consistency, we have discussed this (see response to W1 of Reviewer PWfP) and find that  it is still beneficial for training by providing a more effective comparison signal for policy optimization.

---

### Official Review · Reviewer_PWfP · 2026-03-11

**Soundness:** 3
**Presentation:** 3
**Significance:** 2
**Originality:** 3
**Overall Recommendation:** 4
**Confidence:** 3

**Summary:**

This paper proposes a method for training large language models to form dynamic workflows for Text-to-SQL tasks through reinforcement learning.
The paper models the problem as follows:
Given a request q, the LLM needs to:
1)  Identify available Actors (A)
2)  Select a workflow template (T) based on the query Q
3)  Assign the workflow template to corresponding modules , forming a concrete workflow.

The core innovations declared structurally include two main modules:
1) Dynamic Actor masking: During training, a threshold r is set to allow only r ratio of Actors to be used when constructing the workflow in each iteration (Actors are defined as tools with fixed input-output interfaces).
2) Pseudo-reward mechanism: During training, with a fixed probability p, the following mechanism is used: bypass the execution pipeline that would generate real scores, and instead use an external agent to judge whether the workflow is better or worse compared to a so-called baseline workflow.

**Compliance With Llm Reviewing Policy:**

Affirmed.

**Final Justification:**

The author addressed my main concern in rebuttal, and I will maintain my score.

**Key Questions For Authors:**

1) The consistency between the LLM judge's pseudo-rewards and real execution rewards is not analyzed: Section 4.2.3 uses LLM for relative comparison, replacing real execution rewards with probability p. However, the paper does not analyze how consistent the LLM judge's assessments are with real execution results. For example, do workflows judged as "better" by the LLM actually achieve higher execution accuracy? This analysis is crucial for validating the effectiveness of the pseudo-reward mechanism.
2) LLM evaluation mechanism lacks execution history information: When the paper uses LLM for relative evaluation, it does not pass execution history information, nor does it provide specialized training for the evaluation model. This may lead to evaluations based only on superficial semantic matching, unable to truly reflect the actual execution effectiveness of workflows. The paper does not explain whether this design choice affects the reliability of pseudo-rewards.
3) The theoretical argumentation in Section 3 is largely complete, but there is room for improvement: Appendix B.3 provides a quantitative relationship (lower bound formula) between dynamic advantage Δ and workflow heterogeneity. This result is actually a natural extension of Theorem 3.1 and is directly related to the empirical distance calculation in Appendix C.2. It is recommended to move this core quantitative relationship to the main text to enhance the depth and operational value of the theoretical section. Counterexamples and other details can remain in the appendix.

**Limitations:**

1) Deep understanding of LLM evaluation mechanism: The current mechanism of using LLM as a judge for pseudo-rewards may only be based on superficial semantic matching, lacking deep understanding of Actor-Template matching. If historical messages could be passed or the evaluation model could be specially trained, this might reduce "noise effects" in experiments and improve evaluation accuracy.
2) Pseudo-reward introduction strategy: The paper introduces LLM as a pseudo-reward judge with probability p. This random strategy may be less balanced than a fixed ratio (e.g., every 1/p queries). However, the impact of this design choice requires further analysis.
3) Exploration efficiency of dynamic masking: The current random masking of actors does not consider dependencies between actors or query complexity. For extremely complex queries, random masking may mask necessary actors, leading to ineffective exploration. Structured masking or adaptive masking strategies based on query difficulty could be explored.

**Strengths And Weaknesses:**

Strengths:
1) To reduce verification costs on large datasets, the method adopts LLM-based relative monitoring to evaluate workflow effectiveness scores. This approach positions workflow allocation as the core innovation, improving efficiency through focused examination of workflow allocation.
2) The paper proposes a deterministic exploration mechanism based on action masking. During training, a historical action selection record is maintained. When the reinforcement learning agent attempts to repeatedly select already-verified Actor-Template combinations, dynamic action masking forces it to explore new combinations, effectively preventing the policy from getting stuck in local optima and improving the generalization ability of Text-to-SQL workflows.
3) The reward design employs dense rewards while using methods such as masking some Actor classes to avoid repeated optimal solutions. The paper has validated this claim through relatively comprehensive experiments.

Weaknesses:
1) Regarding the selection of comparison samples for LLM-based relative evaluation: "If the training sample contains a verified real workflow, we directly use it as the baseline; otherwise, we prompt a stronger LLM to generate a baseline workflow using the same input as a comparison reference." This design requires further justification, particularly regarding the difference from directly using a stronger LLM to generate workflows, and the rationale for using a stronger LLM as a baseline solution.
2)  Regarding the use of large language models to evaluate generated results: The evaluation model may need appropriate training and more information transfer regarding execution history. Without specialized training or history transfer, there may be an issue of "being unable to truly reflect execution results, only superficially evaluating whether the workflow template allocation is reasonable based on prompts." Additionally, should different models be used rather than self-evaluation? The paper does not provide detailed explanation on this point.
3) The contribution focuses on dynamic construction at the workflow level, with innovation at the methodological level, but does not propose new solutions to the underlying challenges of the Text-to-SQL task.

---

> ### Author Rebuttal · Authors · 2026-03-31
>
> Thank you for the insightful and detailed comments. Before addressing specific concerns, we provide three clarifications to avoid potential misunderstandings.
>
> **Why Pseudo-rewards?**
>
> During RL training, computing real rewards requires fully executing the rollout workflow, causing significant GPU idle time. We therefore introduce pseudo-rewards to accelerate reward evaluation and reduce training cost, while controlling their influence via the substitution probability *p*.
>
> **How Implemented?**
>
> We clarify that our LLM judge incorporates evaluation principles derived via iterative self-evolution over labeled historical trajectories. In each iteration, the judge evaluates samples under current principles; error cases (FP/FN) are identified and fed back as structured evidence; the LLM then revises the principles to correct observed failure patterns. After N iterations, we select the principles with the best composite score (AA − FPR). Code is available at: https://anonymous.4open.science/r/rebuttal-F6D5/codes/pseudo
>
> **What Affect?**
>
> Pseudo-rewards accelerate both evaluation and training. While they introduce noise, its impact is controllable via p, yielding a trade-off between efficiency and performance. As shown in Table 4, a modest substitution rate (p=0.1) does not hinder learning; instead, it helps avoid local optima and promotes broader policy exploration, even outperforming training with real rewards only.
>
>  ---
>
> We will respond to the weaknesses and questions below, with a slightly reordered presentation for better clarity.
>
> **Response to Q1**
>
> To directly address this concern, we conduct ablation studies to evaluate alignment between pseudo-rewards and real rewards (Table below). Specifically, we measure whether workflows preferred by the LLM judge are indeed better under real execution using "Success Rate".
>
> | Methods                      | Success Rate (%) |
> |-----------------------------|------------------|
> | Qwen-32B (small)            | 50.4             |
> | Qwen-235B (big)             | 50.7             |
> | Qwen-235B + baseline       | 49.3             |
> | Qwen-235B + principles      | 56.9             |
> | Qwen-235B + both            | 57.0             |
> ---
> The results show that `+both` and `+principles`  both improve this consistency. Although `+baseline` reference alone does not noticeably improve consistency, we will discuss later that it is still beneficial for training by providing a more effective comparison signal for policy optimization (see W1).
>
>
> **Response to W2 & Q2**
>
> As noted above, evaluation principles are iteratively refined from historical trajectories and serve as grounded criteria for the LLM judge. The ablation study shows that principles effectively reduce false positives and improve pseudo-reward reliability. We acknowledge that principle construction and pseudo-reward design remain open problems and important directions for future work.
>
>  ---
>
> **Response to W1**
>
> `Point (i): Direct workflow generation by stronger LLMs is infeasible.`
>
> As shown in following Table, stronger LLMs (up to 671B) achieve substantially lower accuracy than our SquRL at much higher latency, indicating that they are insufficient for dynamic workflow construction. The detailed table results can be found at:https://anonymous.4open.science/r/rebuttal-F6D5/assets/llm_baseline.png.
>
> `Point (ii): The stronger LLM as a comparison baseline improves policy exploration.`
>
> Although the baseline does not significantly improve pseudo-reward accuracy, it enhances downstream performance:
>
> | Pseudo Reward Source               | Accuracy |
> |-----------------------------------|----------|
> | Real Reward                       | 51.4     |
> | Random Noise (10%)                | 50.5     |
> | Qwen3-235B + w/o baseline (10%)   | 52.1     |
> | Qwen3-235B + w/ baseline (10%)    | 52.9     |
>
> We attribute this to a shift in the distribution of noisy rewards rather than a reduction in noise. The baseline alters which rollouts receive incorrect rewards: workflows outperforming a competent baseline are more likely to exhibit meaningful structure (e.g., coherent plans for complex queries), even if execution fails. This promotes more effective exploration, consistent with gains in Table 4.
>
> ---
>
> **Response to W3**
>
> We respectfully disagree that our work does not introduce a new solution: SquRL provides a new workflow-level solution for Text-to-SQL. This matters because actor-level methods typically address only isolated sub-challenges, while real Text-to-SQL queries often require multiple capabilities to work together; accordingly, SquRL consistently outperforms strong static pipelines built from the same underlying actors.

---

> > ### Author Rebuttal · Reviewer_PWfP · 2026-04-01
> >
> > The concerns expressed earlier have been resolved. Kindly accommodate the insights in the revised version of the paper. Scores have been increased.

---

### Official Review · Reviewer_GmLr · 2026-03-12

**Soundness:** 3
**Presentation:** 2
**Significance:** 2
**Originality:** 3
**Overall Recommendation:** 4
**Confidence:** 4

**Summary:**

This paper introduces SquRL, a reinforcement learning framework that learns to dynamically construct query-specific workflows for Text-to-SQL rather than applying a single static pipeline to all queries. The system decomposes existing methods into modular actors within the Squrve framework and trains a policy model via SFT followed by GRPO-based RL with execution feedback. The RL stage incorporates dynamic actor masking to prevent policy collapse and pseudo rewards via LLM-as-judge to reduce training cost. An oracle analysis on SynSQL (81.5% accuracy) validates that a per-query workflow gap exists. SquRL-7B achieves 86.27% on Spider-Dev and 67.60% on BIRD-Dev, roughly 2 percentage points above the best static baseline (RSL-SQL), with larger gains on complex queries.

**Compliance With Llm Reviewing Policy:**

Affirmed.

**Key Questions For Authors:**

Q1) For static baselines in Table 1, does "five workflows" mean five runs of the same pipeline? I think providing SquRL single-pass results alongside RSL-SQL with 5x self-consistency would make for a cleaner comparison.

Q2) What is the total computational cost of SquRL training (GPU hours, workflow executions, API calls for pseudo rewards)?

Q3) How does SquRL compare against a simpler routing baseline, for example a lightweight classifier that predicts which static workflow performs best based on query length or schema size?

Q4) What fraction of training queries during SFT had no valid workflow found? I am curious how this affects the RL stage.

**Limitations:**

I did not find a discussion of computational costs, which could be substantial given workflow execution during rollouts. The tight coupling to the framework and its specific actor pool may also limit generalizability.

**Strengths And Weaknesses:**

Strengths:

S1) The problem formulation is well-motivated. The two-question structure (Q1: does a gap exist between static and dynamic workflows, Q2: can we learn a policy to exploit it) provides clear narrative direction, and the oracle analysis on SynSQL offers concrete evidence that the gap is exploitable.

S2) Dynamic actor masking is a practical training mechanism that addresses a real problem. Figure 3 shows that models trained with masking maintain robust performance across retention rates, whereas models without masking degrade sharply when actors become unavailable.

S3) The pseudo reward mechanism provides an interesting cost-accuracy tradeoff. Replacing 10% of execution-based rewards with LLM-based pairwise comparisons improves Spider-Dev accuracy from 82.71% to 85.10% (Table 4), suggesting denser feedback from partially correct workflows provides a useful learning signal.

S4) Results scale consistently across model sizes (1.5B, 3B, 7B) and across all four benchmarks, with SquRL-1.5B already surpassing the best static baseline on Spider-Dev and Spider-Test.


Weaknesses:

W1) I did not find error bars, confidence intervals, or significance tests in the reported results. Given that RL training is sensitive to random seeds and the evaluation uses majority voting over 5 workflows, the reported improvements of roughly 2 percentage points may fall within run-to-run variance.

W2) Table 1's caption states that five workflows are generated per query with majority voting for all methods. However, for static baselines with a fixed pipeline, this is functionally self-consistency rather than workflow diversity. A controlled comparison showing SquRL's single-pass accuracy alongside RSL-SQL with 5x self-consistency would help isolate the contribution of dynamic selection from diverse sampling.

W3) I could not find a discussion of computational cost at training or inference time. SquRL training requires executing workflows during RL rollouts and invoking LLM-as-judge for pseudo rewards, both of which could be expensive. Including GPU hours, API call counts, or latency comparisons would help clarify practical applicability.

W4) While Appendix E provides qualitative case studies of workflow selections before and after training, I think a quantitative analysis of what workflows SquRL selects across different query types, how they compare to oracle choices, and where the policy fails would improve interpretability.

---

> ### Author Rebuttal · Authors · 2026-03-31
>
> We appreciate your thorough review. We would like to clarify some misunderstandings and provide further evidence to substantiate SquRL contributions and superiority.
>
> **Response to W1**
>
> Starting from the same early-stage checkpoint, we run training for 100 steps with 5 random seeds (42/123/777/2026/3407)  under otheriwse identical settings. The resulting accuracy is 50.35 ± 0.61 (std) on the same test set. A paired t-test gives p = 0.447, indicating that the performance variation across seeds is not statistically significant. The detailed experimental results can be found at: https://anonymous.4open.science/r/rebuttal-F6D5/assets/random_seed.png.
>
> ---
>
> **Response to W2 & Q1**
>
> The reviewer is correct that these settings are functionally different: for SquRL, we generate 5 independently constructed workflows, each executed once, rather than repeatedly sampling from the same fixed pipeline. This is the setting reported in Table 1. Accordingly, for static baselines, the 5-sample setting is closer to self-consistency, whereas for SquRL it reflects workflow diversity.
>
> To explicitly separate these factors, we add a controlled comparison on SynSQL, reported in the following table. RSL-SQL improves from 57.42 to 59.29 with 5 executions. SquRL with a single workflow already reaches 61.17—surpassing RSL-SQL with 5× self-consistency—and gains further from more executions (63.35) or multiple workflows (64.42). These results show that SquRL’s gain is not explained by self-consistency alone: dynamic workflow selection is beneficial even in the single-pass setting, and workflow diversity provides an additional complementary improvement.
>
> | Methods                          | Workflows | Executions per WF | Accuracy |
> |----------------------------------|----------|-------------------|----------|
> | RSL-SQL (Adopted in Table 1)     | 1        | 1                 | 57.42    |
> | RSL-SQL                          | 1        | 5                 | 59.29    |
> | SquRL                            | 1        | 1                 | 61.17    |
> | SquRL                            | 1        | 5                 | 63.35    |
> | SquRL (Adopted in Table 1)       | 5        | 1                 | 64.42    |
>
> ---
>
> **Response to W3 & Q2**
>
> We report the training and inference cost of SquRL. During training, each RL step requires 0.124 GPU hours, with ~502K API tokens and 190.22s latency due to workflow execution and LLM-based reward evaluation.
>
> At inference, the cost scales with the number of workflows. SquRL requires 8,038 tokens / 10.51s for a single workflow and 41,783 tokens / 20.82s for 5 workflows. Notably, latency grows sublinearly with the number of workflows due to parallel execution.
>
> ---
>
> **Response to W4**
>
> We thank the reviewer for this suggestion and add a quantitative analysis of workflow selection across query types.
>
> Results show that SquRL learns adaptive workflow complexity conditioned on query difficulty. For simple queries, SquRL uses significantly simpler workflows (1.16 actors, 1.16 length) than the base model (4.24, 1.41), while achieving much higher accuracy (82.06 vs. 58.26) and lower latency (10.71s vs. 29.56s). For complex queries, SquRL selectively increases workflow complexity (2.74 actors, 1.61 length), leading to large accuracy gains (59.47 vs. 23.28) with moderate latency (16.34s vs. 28.87s).
>
> These results indicate that SquRL does not rely on uniformly complex workflows; instead, it learns to allocate complexity where needed, simplifying easy cases while expanding reasoning for harder ones. This adaptive behavior explains both the improved accuracy and efficiency observed after training.
>
> | Methods   | Query Type | Avg. Actors / WF | Avg. Length / WF | Accuracy | Latency |
> |-----------|------------|------------------|------------------|----------|---------|
> | Qwen-7B   | Simple     | 4.24             | 1.41             | 58.26    | 29.56   |
> | Qwen-7B   | Complex    | 5.37             | 1.54             | 23.28    | 28.87   |
> | SquRL-7B  | Simple     | 1.16             | 1.16             | 82.06    | 10.71   |
> | SquRL-7B  | Complex    | 2.74             | 1.61             | 59.47    | 16.34   |
>
> ---
>
> **Response to Q3**
>
> We thank the reviewer for this insightful suggestion. We train a lightweight BERT classifier as a routing baseline. Specifically, we collect data by running 5 static workflows on 1,200 randomly sampled queries, yielding 870 valid instances with at least one correct execution. The classifier takes the question and schema as input and predicts the best workflow.
>
> On the test set, the BERT-classifier achieves 49.48%, substantially lower than SquRL (61.17%). This suggests that simple routing based on input features is insufficient to capture the complex, context-dependent workflow selection required in this setting. In contrast, SquRL learns a policy to construct more effective workflows, yielding substantially better performance.

---

> > ### Author Rebuttal · Reviewer_GmLr · 2026-04-05
> >
> > I thank the authors for their thorough and constructive rebuttal. I appreciate the additional experiments and clarifications regarding seed sensitivity, the distinction between workflow diversity and self-consistency, the training and inference cost, the adaptive workflow analysis, and the routing baseline. Overall, the paper is in better shape after addressing these concerns.
> >
> > Given these improvements, my assessment remains positive. While some limitations still remain, I believe the rebuttal has sufficiently strengthened the submission, and I therefore retain my score and recommendation for acceptance.

---

### Official Review · Reviewer_oHVF · 2026-03-13

**Soundness:** 3
**Presentation:** 4
**Significance:** 3
**Originality:** 3
**Overall Recommendation:** 5
**Confidence:** 5

**Summary:**

The authors assess a pressing question in the Text-to-SQL domain: how to overcome the limitations of rigid, static pipelines that struggle to generalize across complex and out-of-distribution queries. The article attempts to examine a challenge in solving this issue by introducing SquRL, a two-stage learning framework that trains LLMs to dynamically construct task-specific workflows at inference time. By combining an initial supervised fine-tuning phase to learn valid workflow structures with a reinforcement learning phase, optimized via a multi-tiered rule-based reward system, dynamic actor masking for exploration, and LLM-driven pseudo rewards for efficiency, the system adaptively selects the most effective reasoning components (actors) for each specific query. Ultimately, extensive experiments demonstrate that SquRL's dynamic approach consistently outperforms static baselines like DIN-SQL and MAC-SQL across major benchmarks, with accuracy and efficiency gains on highly complex tasks.

**Compliance With Llm Reviewing Policy:**

Affirmed.

**Key Questions For Authors:**

NA

**Strengths And Weaknesses:**

Strength:

- The conceptual shift from static pipelines to dynamic workflows is a brilliant idea that opens entirely new doors for accuracy and efficiency improvements in the Text-to-SQL domain.

- The introduction of dynamic actor masking to enforce exploration during the reinforcement learning phase is a highly novel solution to prevent the policy from collapsing into a small set of dominant workflows.

- The design of the multi-tiered, rule-based reward function is exceptionally well thought out. By evaluating formatting, execution validity, accuracy, and runtime sequentially, the framework provides dense feedback without unnecessary computational overhead.

- The ablation study clearly justifies the architecture, successfully isolating the impact of the Supervised Fine-Tuning and Reinforcement Learning stages and proving that both are critical to the framework's success.

Weaknesses:

- The biggest concern about this work is the baselines. Most of the baseline methods used in this work are relatively old approaches, new methods such as CHASE-SQL, Agentar-Scale-SQL, and AskData + GPT-4o.

- I think there is a similarity between dynamic workflow construction and tool selection and planing for LLM agents. In that sense, I think it would be beneficial to make a comparison with agent planning methods as well.

---

> ### Author Rebuttal · Authors · 2026-03-31
>
> We thank the reviewer for the thoughtful suggestions and constructive feedback.
>
> **Response to Weakness One**
>
> We agree that including comparisons with more recent systems would strengthen the evaluation. In our framework, such methods can be naturally integrated as additional actors, enabling evaluation under the same setting as SquRL. We will incorporate these comparisons to provide a more comprehensive and up-to-date assessment.
>
> ---
>
> **Response to Weakness Two**
>
> This is an important and well-motivated direction. Learning planning models for effective tool selection remains a challenging problem. Compared to such settings, our problem is relatively easier to optimize. In agent-based scenarios, models typically require multi-step tool interactions, where only the final outcome is reliably verifiable, while intermediate steps lack supervision, leading to inherent credit assignment challenges.
>
> A promising direction is to incorporate search-based methods, such as MCTS (e.g., Tree Search for LLM Agent Reinforcement Learning), to better estimate intermediate rewards. However, such approaches often incur substantially higher computational cost and latency. We consider this an important open problem and plan to explore it in future work.

---

### Decision · Program_Chairs · 2026-04-30

**Decision:**

Accept (regular)

**Comment:**

This paper proposes SquRL, a two-stage SFT+RL framework that trains LLMs to dynamically construct query-specific workflows for Text-to-SQL, replacing static pipelines with adaptive reasoning over modular actors. The conceptual shift is well-motivated, backed by an oracle analysis demonstrating the existence of a performance gap, and supported by consistent empirical gains across Spider, BIRD, SynSQL, and Spider2.0-Lite benchmarks at multiple model scales.

Key concerns raised — including statistical significance of improvements, the distinction between workflow diversity and self-consistency, training and inference cost, alignment between pseudo-rewards and execution outcomes, and the potential system-specificity of the learned policy — were addressed in the rebuttal with concrete experiments. The authors are encouraged to incorporate these results into the final version for a stronger version of this paper.